



## Spatial Influence of Fault-Related Stress Perturbations in Northern
## Switzerland
Lalit Sai Aditya Reddy Velagala[1,4], Oliver Heidbach[1,2], Moritz Ziegler[1,3], Karsten Reiter[4], Mojtaba Rajabi[5], Andreas Henk[4], Silvio B. Giger[6], Tobias Hergert[7].
1 GFZ Helmholtz Centre for Geosciences, Telegrafenberg, 14473 Potsdam, Germany.
2 Institute for Applied Geosciences, Technische Universität Berlin, 10587 Berlin, Germany.
3 Professorship of Geothermal Technologies, Technical University Munich, 80333 Munich, Germany.
4 Institute of Applied Geosciences, Technische Universität Darmstadt, 64287 Darmstadt, Germany.
5 School of the Environment, The University of Queensland, QLD, 4072, Australia.
6 National Cooperative for the Disposal of Radioactive Waste, 5430 Wettingen, Switzerland.
7 Institute of Applied Geosciences, Karlsruhe Institute of Technology, 76131 Karlsruhe, Germany.
## Correspondence
Lalit Sai Aditya Reddy Velagala: vlsar24@gfz.de; vlsadityareddy@gmail.com
Oliver Heidbach: heidbach@gfz.de
## Abstract
The spatial influence of faults on the crustal stress field remains a topic of active debate. While it is well
documented that faults often cause perturbations in the stress field at a meter scale, their lateral influence over
greater distances, from a few hundred meters to several kilometers, remains poorly understood. This knowledge
gap largely results from the lateral resolution limit of stress data. To address this, we use a 3D geomechanical
numerical model based on 3D seismic data from northern Switzerland. The model is calibrated with 45 high-
quality horizontal stress magnitude data obtained from micro-hydraulic fracturing (MHF) and sleeve re-opening
(SR) tests conducted in two boreholes in the Zürich Nordost (ZNO) siting region. The 3D seismic and stress data
were collected as a part of site characterization for a potential Deep Geological Repository (DGR) for radioactive
waste. This 3D geomechanical numerical model serves as the reference model in our study and includes seven
faults, implemented as contact surfaces with Coulomb friction. It is then systematically compared to three fault
agnostic models i.e., models without any implemented faults. These fault agnostic models use identical rock
properties and model input parameters, are calibrated with the same 45 horizontal stress magnitude dataset
and have the same model extent, but differ in their discretization and mechanical properties' assignment
procedure. The results show that at distances of < 1 km from faults, differences in maximum horizontal stress
orientation between models range from 3°–6°, and horizontal stress magnitude differences are about 1–2 MPa.
Beyond 1 km distance, the differences reduce to < 1.5° and < 0.5 MPa, respectively. These stress differences are
far smaller than the uncertainties associated with the horizontal stress magnitude measurements at the ZNO
siting region, which average to ±0.7 MPa for the minimum horizontal stress magnitude and ±3.5 MPa for the
maximum horizontal stress magnitude. An important implication of this lateral quantification of fault influence
on stress state is that explicit representation of faults may not be necessary in geomechanical models predicting
the stress state of rock volumes located a kilometer or more from major active faults, an important prerequisite
for any DGR campaign. This structural simplification allows for faster model set-up and discretization, leading to
a significant reduction in the set-up phase and computational time by more than one order, without
compromising the reliability of stress field predictions.
## Short Summary
We assess the fault impact on the stress field in northern Switzerland using 3D geomechanical models, calibrated
with stress data. We see that faults affect the stresses only locally, with negligible impact beyond 1 km,
suggesting that faults may not be necessary in reservoir-scale models predicting stresses of undisturbed rock
volumes, such as for a deep geological repository. Omitting them can substantially reduce modelling time and
computational cost without compromising prediction accuracy.



## 1. Introduction

Characterizing the crustal stress field is essential for understanding both global and local tectonic deformation processes. On a large scale, it provides insights into plate tectonics (Richardson et al., 1979; Cloetingh and Wortel, 1985) and earthquake mechanics (Sibson, 1992; Sibson et al., 2011; Brodsky et al., 2020), while on a local scale, it plays a critical role in the safe planning of many subsurface applications, including subsurface oil and gas exploration and storage (Zoback, 2007; Berard et al., 2008; Fischer and Henk, 2013), geothermal exploration (Catalli et al., 2013; Schoenball et al., 2014; Azzola et al., 2019) and deep geological repositories for nuclear waste (Long and Ewing, 2004; Gens et al., 2009; Jo et al., 2019). The present day stress state also significantly impacts wellbore stability and trajectory optimization, reducing risks and improving drilling operations (Kingsborough et al., 1991; Henk, 2005; Rajabi et al., 2016). Moreover, knowledge of the regional and local stress field aids in assessing seismic hazards and understanding the potential reactivation or generation of faults (Zakharova and Goldberg, 2014; Seithel et al., 2019; Vadacca et al., 2021).

The stress state at a point is described by the Cauchy stress tensor, a symmetric second-order tensor with six independent components. This tensor can be transformed into the principal stress system, where only three mutually perpendicular normal stresses, known as the principal stresses ($S_1$ = maximum principal stress; $S_2$ = intermediate principal stress and $S_3$ is the minimum principal stress), remain and the shear stresses are zero. In reservoir geomechanics, where the target area is the upper crust, it is typically assumed that the principal stresses are the vertical stress ($S_V$), the maximum horizontal stress ($S_{Hmax}$) and minimum horizontal stress ($S_{hmin}$). Based on this, the reduced stress tensor is established by four key parameters: the magnitudes of $S_V$, $S_{Hmax}$, and $S_{hmin}$, and the orientation of $S_{Hmax}$ (Jaeger et al., 2007; Zoback, 2007).

The $S_{Hmax}$ orientation is the most widely available, systematically documented and freely accessible reduced stress tensor component, compiled in publicly available database of the World Stress Map project (Heidbach et al., 2018; Heidbach et al., 2025a). Analyzing the patterns of the $S_{Hmax}$ orientation shows consistent trends over hundreds of kilometers in intra-continental areas, primarily driven by first-order plate tectonic forces and second-order buoyancy forces (Zoback et al., 1989; Zoback, 1992; Heidbach et al., 2018). At the same time, in some regions, significant rotations exceeding 30° are observed on spatial scales ranging from a few tens to a few hundreds of kilometers. It is hypothesized that these variations in $S_{Hmax}$ orientations arise from third-order sources, mainly the active faults (Zoback et al., 1987; Yale, 2003; Heidbach et al., 2007; Tingay et al., 2009; Rajabi et al., 2017b).

A common approach to understand the fault impact on the stress field is to visually interpret laterally scattered $S_{Hmax}$ orientation data. This often leads to attributing the observed variability in $S_{Hmax}$ orientation to the faults present within their respective study areas (Yale et al., 1994; Bell, 1996b; Yale, 2003; Aleksandrowski et al., 1992). While these studies are often convincing, they face two key issues: First, even in areas with relatively high data coverage, such as northern Switzerland (Heidbach et al., 2025a; Heidbach et al., 2025b), and the northern Bowen Basin (Rajabi et al., 2024; Heidbach et al., 2025a), the usable publicly available data records and their resolution are fairly low, with on average approximately about one data record per 138 km² lateral spatial distance, and one data record per 80 km² lateral spatial distance respectively. Second, individual $S_{Hmax}$ orientations usually have an average standard deviation of ±15° (A-Quality) to ±25° (C-Quality), as defined in the World Stress Map (Heidbach et al., 2025a). Together, these issues make it difficult to attribute with confidence the small perturbations in the stress rotations to the faults, especially at spatial scales of 0.1–10 km.

Notable studies from regions with a comprehensive $S_{Hmax}$ orientation dataset show that large-scale faulting does not necessarily result in abrupt $S_{Hmax}$ orientation rotations over continental (> 500 km) and regional scales (100–500 km). For instance, in eastern Australia, the $S_{Hmax}$ orientation rotates smoothly, by up to 50° over less than 100 km despite varying dip and strikes of the major fault systems, from northern Bowen Basin to southern Bowen and Surat basins (Brooke-Barnett et al., 2015; Rajabi et al., 2024) (Fig. 1A, B). However, in the adjacent Clarence-Moreton Basin, rotation of $S_{Hmax}$ orientations is prominent and abrupt when viewed in conjugation with the faults (Rajabi et al., 2017b; Rajabi et al., 2017c; Tavener et al., 2017; Mukherjee et al., 2020) (Fig. 1A, B). Comparable conflicting trends have been reported in other studies as well (Bell and Gough, 1979; Gough and Bell, 1982; Bell and Grasby, 2012), suggesting that the influence of fault systems on $S_{Hmax}$ orientation rotations at continental and regional scale is not straightforward and often not resolvable without ambiguity.





The stress maps typically display an average of all the S$_{Hmax}$ orientation along the length of a borehole and does
not capture potential changes in S$_{Hmax}$ orientation with depth due to interaction with the faults. At borehole scale
studies, distinct variations in S$_{Hmax}$ orientation have been observed vertically on a spatial scale of a few meters.
For instance, Fig. 1D shows an image log of a borehole from the Clarence-Moreton Basin, where S$_{Hmax}$ orientation
abruptly changes by 90° when the borehole intersects a fault. In the San Andreas Fault Observatory Drilling
Borehole, borehole breakouts (BO) and drilling induced tensile fractures (DITF) indicate a change in S$_{Hmax}$
orientation from 25° ± 10° at 1000–1500 m (true vertical depth; t.v.d) to 70° ± 14° at 2050–2200 m (t.v.d) (Chéry
et al., 2004; Hickman and Zoback, 2004; Boness and Zoback, 2006; Zoback et al., 2011). In the KTB drilling
program, S$_{Hmax}$ orientation remained consistent with the regional tectonic-induced patterns except at a depth of
7200 m (t.v.d), where a major fault zone caused a localized reorientation by about 60°, confined to only a few
meters above and below the fault (Brudy et al., 1993; Barton and Zoback, 1994; Brudy et al., 1997). Similar
localized stress reorientations near fault zones and pre-existing fractures have been reported in other boreholes
(Ando, 2001; Tsukahara et al., 2001; Lin et al., 2010; Nie et al., 2013; Cui et al., 2014; Jo et al., 2019; Massiot et
al., 2019; Rajabi et al., 2022; Li et al., 2025). However, borehole-scale studies are generally conducted in vertical
wells and do not capture the potential lateral variations in stress caused by faults. Therefore, it remains unclear
whether these localized findings can be directly extrapolated to explain stress field variations at larger spatial
scales away from the fault zone. This leads to a significant knowledge gap regarding fault's influence on stress
field variations at reservoir scale (Fig. 1C), a scale particularly important for many subsurface applications.
The major challenge for studies focusing on stress field predictions at reservoir spatial scales is the scarcity of
stress magnitude measurements, which makes geomechanical numerical modeling the most effective and often
the only viable approach for predicting the variations in the stress field at this scale. Over the past few decades,
2D and 3D geomechanical numerical models have been developed for this purpose (Henk, 2009, 2020; Treffeisen
and Henk, 2020). These can broadly be grouped into three categories: 1) site-specific models without fault
representation (Lecampion and Lei, 2010; Rajabi et al., 2017c; Ahlers et al., 2021), 2) site-specific models that
include faults but are not explicitly focused on assessing influence of faults on the predicted stress (Reiter and
Heidbach, 2014; Hergert et al., 2015; Bérard and Desroches, 2021) and 3) generic models (Homberg et al., 1997;
Su and Stephansson, 1999; Reiter et al., 2024; Ziegler et al., 2024). While models without faults are
understandably not suitable for evaluating fault-related stress perturbations, the latter two categories often
have limited or no access to reliable in situ stress magnitude data. This hinders their ability to reliably represent
fault-related stress variations in real-world scenarios, as seen in studies by Ziegler et al. (2016) and Hergert and
Heidbach (2011). The necessity to include faults in the models also could not be meaningfully addressed,
especially if the model aims to predict the stress field within an intact and undisturbed rock volume, located
away from active faults.

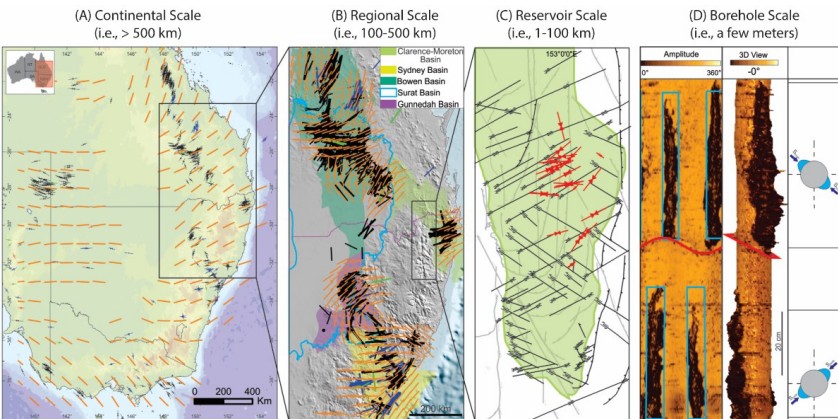


Figure 1: S$_{Hmax}$ orientation stress maps from eastern Australia at A) Continental Scale; B) Regional Scale; C) Reservoir Scale and D) Borehole Scale. On continental
and regional scales, visual observations suggest that faults may have differing influences, as seen in the uniform stress orientation across eastern Australia despite
the presence of faults. However, on a borehole scale, faults can cause local perturbations, evident in the shift of borehole breakout orientations (blue box), which
reflect stress variations across the fault (red line). While research primarily focuses on these three scales, studies examining reservoir scales are scarce due to
lack of reliable stress magnitude data, making it challenging to quantify the spatial influence of faults on the reduced stress tensor components (Image adopted
from Rajabi et al. (2017c)).



In our study, we use 45 reliable and robust stress magnitudes data records, obtained from two deep boreholes,
Trüllikon (TRU1-1) and MAR1-1 (Marthalen), using microhydraulic fracturing (MHF) and dry sleeve re-opening
(SR) test (Desroches et al., 2021a; Desroches et al., 2021b; Desroches et al., 2023) to calibrate 3D geomechanical
numerical models of the Zürich Nordost (ZNO) siting region, northern Switzerland (Fig. 2). The data records were
collected during a comprehensive seismic and drilling campaign to support site selection for a deep geological
repository of radioactive waste (Nagra, 2024b, a). Four variants of the 3D geomechanical numerical model of the
siting region, each with lateral dimensions of 14.7 km × 14.8 km, and a vertical depth of 2.5 km (below sea level;
b.s.l) are used within this study. All models use identical mechanical properties and the same representation of
geomechanically relevant subsurface units. One of the models includes seven contact surfaces with assigned
Coulomb friction representing faults, and serves as the reference model (REF model) (Nagra, 2024c, b). By
systematically comparing the predicted stress fields across all the models, we illustrate the observed
perturbations in the stress field with respect to the reference model and quantify the spatial extent of the stress
perturbations caused by faults.

## 2. 3D Geomechanical Numerical Model with Fault Representation

### 2.1 Geological Background and Model Geometry

The ZNO study area is located in the northern Alpine Foreland of northern Switzerland, approximately 30 km
NNE of Zurich (Fig. 2). It is close to the Black Forest in SW Germany, where pre-Mesozoic basement rocks locally
outcrop (Nagra, 1984, 2002a). The geological evolution of this region was influenced by the development of a
WSW–ENE striking Permo-Carboniferous basin (Gorin et al., 1993; Mccann et al., 2006; Nagra, 2014), formed in
response to the Variscan orogeny and subsequent post-orogenic transtensional processes (Nagra, 1991;
Marchant et al., 2005).
During the Mesozoic, a sequence of sedimentary successions was deposited on the top of the Variscan basement.
This depositional process was prominent especially from the Early to Middle Jurassic due to a combination of
regional tectonic subsidence and sea level changes (Coward and Dietrich, 1989; Nagra, 2024b). The sedimentary
rocks were originally deposited directly on the ocean floor as a result of the landmass corresponding to the
present day Northern Switzerland being submerged in a broad and shallow epicontinental marine setting
(Jordan, 2008; Reisdorf et al., 2011). The Opalinus Clay formation, deposited during the Jurassic Period of the
Mesozoic Era, is of particular importance as it has been selected as the host rock for Switzerland's DGR. Factors
contributing to the effectiveness of Opalinus Clay as a long-term geological barrier are its favorable mineralogy
and associated low permeability, and good sorption and self-sealing properties (Nagra, 2001, 2002b, 2008).
At late Cretaceous and onset of the Cenozoic, the Alpine orogeny, formed by the collision of Adriatic and Eurasian
tectonic plates, led to a significant tectonic activity in the European northern Alpine Foreland (Illies, 1972; Schmid
et al., 1996; Schmid et al., 1997; Cloetingh et al., 2006). This resulted in the formation of basement-rooted, NNE-
striking normal faults, forming the Upper Rhine Valley in combination with the uplift of the Black Forest and
Vosges Mountain Massifs. The formation of the flexural Molasse Basin during the Late Oligocene to Early
Miocene is a result of the downbending of the European plate, in response to the orogenic loading of the Alps,
caused a gentle north-south dip in the Mesozoic strata (Sinclair and Allen, 1992; Kempf and Adrian, 2004;
Sommaruga et al., 2012). In our study area, the Mesozoic strata gently dips SSE (Fig. 3). In the Late Miocene,
continued Alpine deformation propagated into the Northern Foreland, resulting in the formation of the Jura
Mountains and their associated fold-and-thrust belt, primarily further to the west, and reactivating the pre-
existing basement structures (Diebold and Noack, 1997; Burkhard and Sommaruga, 1998; Laubscher, 2010).
These tectonic processes, along with the glacial-interglacial cycles during Pleistocene (Fiebig and Preusser, 2008;
Preusser et al., 2011), have established the present day geological and stratigraphic setting in the region.
The reference model (REF Model) is rectangular, spanning 14.7 km E-W × 14.8 km N-S laterally, and extending to
a depth of 2.5 km below sea level (b.s.l). The upper boundary is defined by the local topography. In the siting
area, $S_{Hmax}$ orientation is 170° ± 11°, in agreement with the regional trend (Nagra, 2013; Heidbach et al., 2025b).
To align the model geometry with the $S_{Hmax}$ orientation, the entire model domain is rotated by 10°
counterclockwise from geographic north, such that its sides are parallel and perpendicular to the mean $S_{Hmax}$
orientation (Fig. 2).



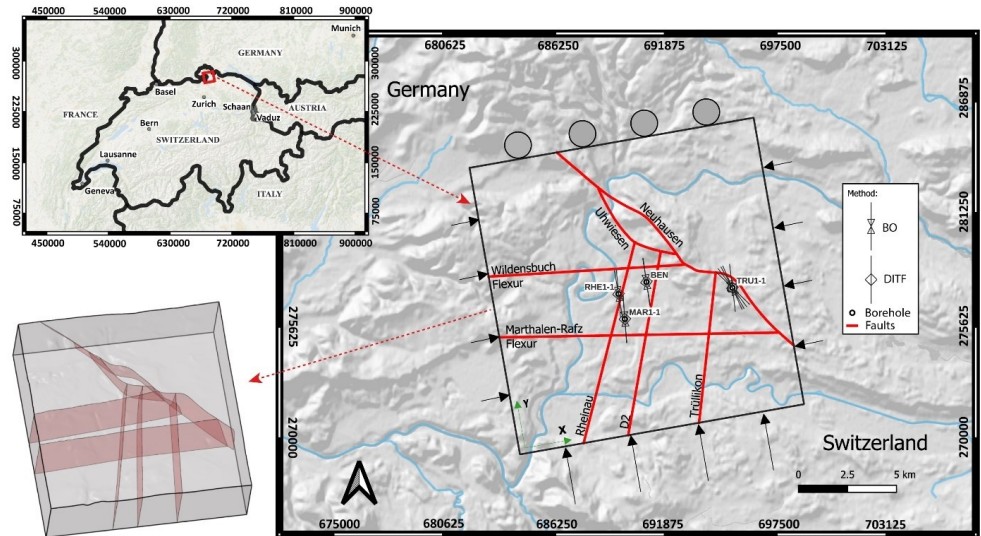


Figure 2: The geographical location and the model boundaries of the ZNO siting region. The red lines represent the surface trace of the faults and flexures, interpreted from the seismic sections of the siting region and extrapolated to the surface. The location of the boreholes Trüllikon (TRU1-1), Benken (BEN), MAR1-1 (Marthalen) and Rheinau (RHE1-1) are shown, along with the $S_{Hmax}$ orientation data records from each borehole. The model is rotated by 10° anticlockwise according to the regional $S_{Hmax}$ orientation values. The black arrows on the sides of the model are the displacement boundary conditions/compression applied on the model boundaries, where the length of the arrows is proportional to the magnitude of the displacement applied. The grey circles on the north of the model indicate that the displacements are constrained perpendicular to this boundary. The co-ordinate reference system used is CH1903/LV03. The insert at the bottom left is the 3D view of the faults (rosa) within the model geometry (grey box).

The present day geomechanically relevant layers were constructed using SKUA-GOCAD v19 software. Successive lithologies with comparable mechanical properties were combined (Table 1). Eventually, the REF model consists of 14 geomechanically different units (Fig. 3). A total of seven faults and flexures, named Neuhausen, Uhwiesen, Wildensbuch, Marthalen-Rafz Flexure, Rheinau, D2, and Trüllikon, were implemented in the model (Fig. 2). These structures are modeled as contact surfaces, weakly interpreted from the regional 3D seismic sections, and are highly simplified for ease of implementation in the model. Here, simplification means merging much smaller segments on 3D seismics into larger, continuous fault planes (Nagra, 2024a) (Fig. 2, 3). Among the faults and flexures, Neuhausen and Uhwiesen dip at 60° toward the northeast, while the others are vertical. Neuhausen is the only fault that displays a stratigraphic offset, with a vertical displacement of approximately 50 m at the base of the Mesozoic units that decreases towards the surface (Nagra, 2002a, 2008, 2024c).

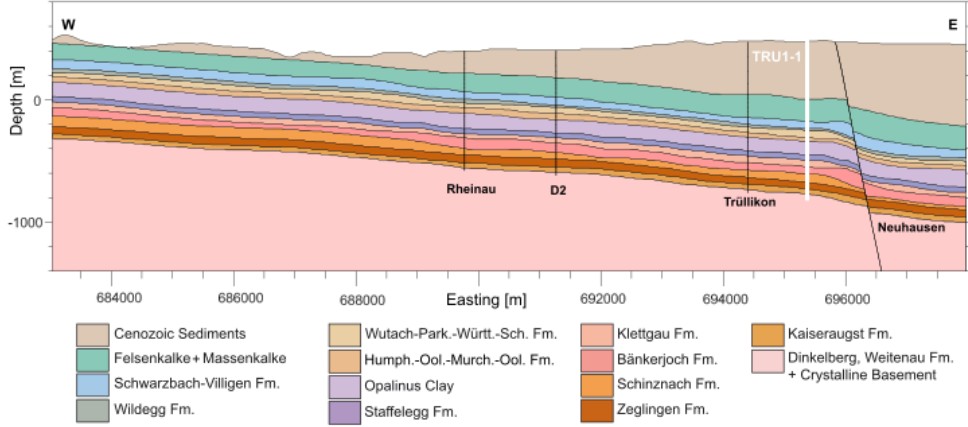


Figure 3: Cross-section of the geomechanical units passing through the Trüllikon borehole (Bold white line, TRU1-1) and a constant northing = 277548 m within the REF model domain. The depth is referenced to sea level. The model includes 14 geomechanical units that exhibit a gentle W-E dip in this cross-section. No



stratigraphic offset is observed across the faults, except the Neuhausen fault, which displays a vertical stratigraphic offset of approximately 50 m. Vertical
exaggeration by a factor of 2.5 is applied to enhance the visibility of thin layers, such as the Wildegg Formation. The respective mechanical properties are shown
in Table 1. Only depths down to −1400 m (b.s.l) are shown for clarity, although the REF model extends to −2500 m (b.s.l). The co-ordinate reference system used
is CH1903/LV03.

## 209    2.2 Reference Model (REF) setup

### 210    2.2.1 Model Assumptions

The primary objective of the REF model is to reliably predict the present day in situ stress state within the ZNO
siting region, using the rock properties and stress magnitude data obtained from deep borehole drilling. To
achieve this, two key simplifying assumptions are made. First, transient effects such as time-dependent tectonic
deformation, or human-induced changes can be neglected while considering only the stress contributions from
the gravitational and tectonic forces. Since the model focuses on static stress field prediction, the rock volume is
assumed to not undergo any transient deformation. Second, linear isotropic elasticity is assumed in the
geomechanical units within the rock volume. This assumption simplifies the required material parameters to
explain the behavior of the rock under stress to just the Young's modulus (E), Poisson's ratio ($\nu$), and density ($\rho$)
of each geomechanical unit (Brandes and Tanner, 2020). The equilibrium condition between the gravitational
and the tectonic forces is governed by a second-order partial differential equation (PDE), with displacement as
the field variable (Jaeger et al., 2007). Since this PDE cannot be solved analytically, a numerical solution approach
is needed and for this, we use the Finite Element Method (FEM). FEM allows use of unstructured meshes to
represent the model volume, which is particularly useful when modeling complex geological features, and
variations in material properties (Mao, 2005; Henk, 2009; Zienkiewicz et al., 2013).

### 225    2.2.2 Model Discretization

The model setup follows a standard series of steps, previously used in other regional geomechanical studies
(Buchmann and Connolly, 2007; Reiter and Heidbach, 2014; Hergert et al., 2015; Ziegler et al., 2016; Rajabi et al.,
2017a). The model volume is discretized into 3D elements, collectively referred to as a mesh. The 3D element
resolution plays a significant role in capturing predicted stress variations, where smaller elements capture a
higher spatial resolution but at increased computational cost (Ahlers et al., 2021; Ahlers et al., 2022). To ensure
a reasonably accurate representation of each geomechanical unit, a minimum of three finite elements are used
in the vertical direction. Accordingly, the top 13 geomechanical units, which are relatively thin (Fig. 3), are
discretized with smaller element sizes vertically, whereas the deeper and thicker Basement unit is represented
with larger element sizes in the vertical direction. A total of 1,923,139 finite elements are used, providing a high-
resolution representation of the geomechanical units, with model resolutions varying from 100-150 m laterally
and 5-20 m vertically. We use first-order elements in this study, with linear shape functions, and the discretization
is done using Altair HyperMesh 2023.1 software package.

### 238    2.2.3 Mechanical Rock properties and Fault properties.

Geological units, with similar mechanical properties, are grouped into the same geomechanical unit for simplicity.
Each element in the mesh is assigned mechanical properties based on the corresponding geomechanical unit.
The mechanical properties E [GPa], $\nu$ [-], and $\rho$ [kg/m³], used in the models are derived from core tests and
petrophysical logs obtained from the TRU1-1 and MAR1-1 boreholes. From the range of values for each
geomechanical unit, the median values (P50) are used for the model, summarized in Table 1. Geological faults
are implemented as contact surfaces that can slip under mechanical loading as a structural response to stress
conditions, depending on their friction properties. In the REF model, contact surfaces are assigned a friction
coefficient of 1 and a zero cohesion, values chosen to best represent the fault properties in the region (Nagra,
2024b).




Table 1: Different geological formations with respective mechanical properties. The abbreviations are used solely to indicate the respective formations in the figures of this paper. Geological formations with similar geomechanical properties are aggregated together in the 3D geomechanical numerical models and are referred to as geomechanical units throughout the paper. Throughout the rest of this paper, the respective units can also be matched with the corresponding colors shown in Fig. 3 and to the abbreviations given here.

| System | Group | Formation | Abbreviation used | $\rho$ [kg/m³] | $\nu$ [−] | E [GPa] |
|---|---|---|---|---|---|---|
| Quaternary, Paleogene and Neogene | | Cenozoic Sediments | CeSe | 2350 | 0.30 | 15 |
| Jurassic | Malm | «Felsenkalke» + «Massenkalk» | MaFeMa | 2685 | 0.18 | 31 |
| | | Schwarzbach-Villigen Fm. | MaScVi | 2685 | 0.20 | 40 |
| | | Wildegg Fm. | MaWi | 2610 | 0.26 | 18 |
| | Dogger | Wutach Fm. | DoWuVaPa | 2530 | 0.32 | 13 |
| | | Variansmergel Fm. | | | | |
| | | «Parkinsoni-Wüttembergica-Sch. » | | | | |
| | | «Humphriesoolith Fm. » | DoHuWeMu | 2540 | 0.28 | 14 |
| | | Wedelsandstein Fm. | | | | |
| | | «Murchisonae-Oolith Fm.» | | | | |
| | | Opalinus Clay Fm. | DoOp | 2520 | 0.37 | 11 |
| | Lias | Staffelegg Fm. | LiSt | 2540 | 0.26 | 18 |
| Triassic | Keuper | Klettgau Fm. | KeKl | 2570 | 0.23 | 17 |
| | | Bänkerjoch Fm. | KeBä | 2700 | 0.22 | 23 |
| | Muschelkalk | Schinznach Fm. | MuSc | 2710 | 0.24 | 32 |
| | | Zeglingen Fm. | MuZe | 2840 | 0.19 | 36 |
| | | Kaiseraugst Fm. | MuKa | 2620 | 0.30 | 23 |
| | Bundsandstein | Dinkelberg Fm. | DiWeCr | 2540 | 0.27 | 34 |
| Permian | Rotliegend | Weitenau Fm. | | | | |
| Crystalline Basement | | Crystalline basement. | | | | |

## 2.2.4 Model Calibration

The present day stress state is computed by applying vertical loading simulating the gravitational forces and lateral displacement boundary conditions to simulate the tectonic loading from the geological history. These boundary conditions are chosen so that the modeled stresses best fit to the stress magnitude data, a process known as model calibration (Reiter and Heidbach, 2014; Ziegler and Heidbach, 2020).

The horizontal stress magnitude data are originally determined as ranges but the mean of these ranges was used for the model calibration. The $S_{hmin}$ magnitude ranges (Fig. 5: red bars) are derived from the micro-hydraulic fracturing (MHF) tests and dry sleeve reopening (SR) tests (Desroches et al., 2021a; Desroches et al., 2021b; Desroches et al., 2023; Nagra, 2024c) provide the basis to bracket the ranges for the $S_{Hmax}$ magnitudes (Fig. 5: blue bars).

The model is calibrated with 30 $S_{hmin}$ and 15 $S_{Hmax}$ magnitudes (Fig. 5). It is done using the PyFast Calibration tool (Ziegler and Heidbach, 2021), which uses a linear regression-based algorithm to compute the best-fit lateral displacement boundary conditions by minimizing the differences between the modeled and measured stress magnitudes. To achieve the best fit of the boundary conditions, a total perpendicular displacement of 0.82 m is applied in the east–west direction and 4.2 m in the south, both shortening the model volume, while the northern boundary remains fixed (Fig. 2). Displacements parallel to the boundaries are permitted on all lateral faces. At the base, vertical displacement is constrained, while horizontal displacement is permitted; the model top remains fully unconstrained. The numerical solution is computed using the Simulia Abaqus V2021 finite element solver. The results are analyzed using Tecplot 360 EX 2023 R2 along with the Geostress V2.0 add-on library (Stromeyer et al., 2020).



## 3. Model set-up of 3D Geomechanical Numerical Models without Fault Representation

### 3.1 Model discretization Strategies

Removing the fault implementation from the 3D models helps us use different model discretization strategies. Two different model discretization strategies were used to set-up three additional fault agnostic 3D geomechanical numerical models.

The standard procedure discretizes each geomechanical unit individually using the definition of its top and bottom interface surfaces. Each element of the unit is assigned to the appropriate mechanical properties (Fig. 4A) directly from the stratigraphic definition. While this approach results in a smooth unit boundary, it requires substantial manual effort and is particularly time-consuming when working with models containing many geomechanical units.

In order to simplify the setup and discretization procedure of fault agnostic models, we use ApplePy (Automatic Partitioning Preventing Lengthy Manual Element Assignment), a Python-based tool that automates the discretization and property assignment process (Ziegler et al., 2020). The entire model volume is discretized first as a largely homogeneous mesh, ignoring both lithological interfaces and fault structures. ApplePy uses the depth values of the stratigraphic boundaries to decide which element belongs to which lithological unit/geomechanical unit (Fig. 4B). Although this approach introduces step-like transitions at unit boundaries which looks optically unrealistic, it significantly reduces manual meshing time, especially for large or complex models, like the REF model without compromising the stress prediction capability of the final 3D geomechanical numerical models.

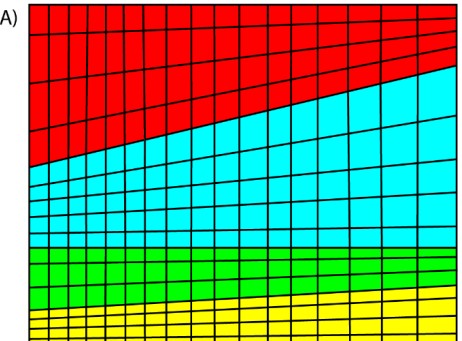 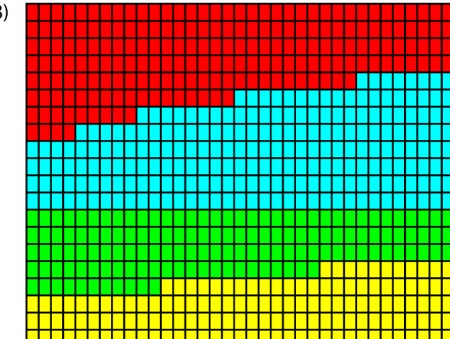

Figure 4: Visual comparison of A) the standard procedure and B) the ApplePy procedure for discretization and mechanical property assignment to geomechanical units. The four colors represent distinct geomechanical units, each with unique lithologies and mechanical properties. In the standard procedure, each geomechanical unit is discretized individually and later connected to each other by matching the nodes along the common interfaces. The resulting geomechanical units' interfaces are smoother. The ApplePy procedure is significantly faster by approximately an order. Here, the whole model volume is discretized in a single step ignoring the interfaces. Due to the working principle of ApplePy, a step-like transitions at unit boundaries are observed.

### 3.2 Model Realizations and Configurations

The three fault agnostic 3D geomechanical numerical models follow the general model workflow of the REF model i.e., the model parameterization and calibration are the same (Sect. 2.2), along with similar model extents (Sect. 2.1). The only differences lie in the model discretization strategies (Sect. 3.1) and resolutions. Out of these three models, one is set up using the standard procedure and two are set up using the ApplePy procedure. Table 2 presents the technical details on the number of elements and spatial resolution of each model used, along with the corresponding best-fit displacement boundary conditions obtained after applying FAST Calibration tool. The brief description of the three models without faults structures are:

- REF-NF Model: REF-NF model is directly derived from the REF model, maintaining identical geometry and mechanical property assignments. The only difference between this model and the REF model is that faults are omitted. This means for the six faults except the Neuhausen Fault that the contact surfaces are eliminated and double nodes on opposite sides of the former faults are equivalenced. For



the Neuhausen Fault, this procedure is not possible due to the lithological vertical 50 m offset which is
represented in the mesh. To prevent slip along this surface, the fault's friction coefficient is artificially
increased to 50.
• AP Model: The AP model maintains the same extents and mechanical properties as the REF and REF-NF
models but uses a modified discretization, not tracking geological interfaces. Property assignment to
the elements is done using the ApplePy tool. It does not incorporate faults, eliminating the need for
contact surfaces within the model framework and has approximately 50% more elements than the REF
and REF-NF models.
• AP-H model: The AP-H model is a higher resolution version of the AP model, with twice the number of
elements. All the other features of the model are the same as the AP model.
Table 2: Summary of technical specifications for all model realizations used in this study. To ensure adequate numerical representation in the ApplePy models
(AP and AP-H models), each geomechanical unit layer is modeled with at least three elements vertically, with a higher resolution allocated to the Mesozoic and
Cenozoic units of interest compared to the basement. The REF-NF, AP, and AP-H models have no fault representation. The listed vertical resolution values apply
only to the Mesozoic units, as these are the target for planning the DGR facility. Vertical resolution values for ApplePy models are approximate, as they vary by
geomechanical unit with depth.

| Model Realization | Fault-Representation | Discretization Procedure | Number of Elements | Vertical Resolution of the Mesozoic Elements [m] | Lateral Resolution [m] | Displacement Boundary Conditions | |
|---|---|---|---|---|---|---|---|
| | | | | | | South [m] | East-West [m] |
| REF Model | Yes: Present as contact surfaces | Standard procedure | 1,923,139 | 5-20 | 100–150 | 4.1 | 0.82 |
| REF-NF Model | No: Directly deleted and mechanically disabled with high friction coefficient of 50 | | 1,923,139 | 5-20 | 100–150 | 4.2 | 0.90 |
| AP Model | No: Directly Excluded | ApplePy procedure | 2,826,240 | ~7 (non-basement units) | 80–110 | 4.23 | 0.93 |
| AP-H Model | | | 5,974,150 | ~4 (non-basement units) | 60–80 | 4.25 | 0.90 |

## 329 4. Results

### 330 4.1 1-D results of the horizontal stress magnitudes along the borehole trajectories

The resulting predicted horizontal stress magnitudes from all the model realizations are presented together with
the measured $S_{hmin}$ (red bars) and $S_{Hmax}$ (blue bars) magnitude ranges along the TRU1-1 and MAR1-1 borehole
trajectories in Fig. 5. In our study, since the fault agnostic models are compared against the REF model, we first
look at the results of the REF model in isolation before examining the results from all the four model realizations
together.
In general, the predicted horizontal stress magnitude from the REF model (Fig. 5; vertical red line changing with
depth), align reasonably well with the measured stress ranges across different geomechanical units. However,
some discrepancies are present, particularly in the Klettgau and Bänkerjoch formations, where the REF model
underestimates $S_{hmin}$ magnitudes, and in the Schinznach formation, where it overestimates $S_{hmin}$ magnitudes.
These deviations arise because the REF model uses P50 (median) stiffness values for stress simulations, whereas
the MHF are representative of rock volume at a meter scale. Also, for the model calibration with the measured
horizontal stress magnitudes, the REF model uses P50 (median) horizontal stress magnitude values in spite of the
MHF tests resulting in ranges (red and blue bars in Fig. 5). Therefore, the stress predictions may vary from the
assumed P50 value at a particular point in the subsurface.





Figure 5: Measured and modelled $S_{hmin}$ magnitude and $S_{Hmax}$ magnitude ranges of all the model realizations with depth (t.v.d) along the TRU1-1 (top row) and
MAR1-1 (bottom row) borehole. The horizontal red bars represent the lower-upper ranges of the $S_{hmin}$ magnitude and horizontal blue bars represent the lower-
upper ranges the $S_{Hmax}$ magnitudes (Nagra, 2024c, b). The geomechanical units are represented by their respective colors and abbreviations, consistent with Fig.
3 and Table 1.



The predicted results from all the model realizations, regardless of fault implementation or exclusion, also align
well with the measured horizontal stress magnitudes ranges along both the borehole trajectories across different
geomechanical units and also with respect to the REF model. Little but negligible differences of < 1 MPa in the
$S_{Hmax}$ stress magnitudes can be found at ~475 m (t.v.d) along the TRU1-1 borehole and at ~250 m (t.v.d) along
the MAR1-1 borehole in the AP and AP-H models (Fig. 5). This is likely due to a high stiffness contrast between
the Cenozoic Sediments (E = 15 GPa) and Felsenkalke + Massenkalke (E = 31 GPa) units, the transition boundary
of which is differently discretized due to ApplePy usage. Another such difference can be found at the Zeglingen
Fm. (E = 36 GPa), Kaiseraugst Fm. (E = 23 GPa) and the Dinkelberg, Weitenau Fm. and Crystalline basement (E =
34 GPa), which is also due to the widely varying stiffness contrasts. While the P50 values of the horizontal stress
magnitudes fit well across all the predicted horizontal stress magnitudes, local deviations occur due to presence
of geomechanical anomalies. For instance, stress magnitude data at 916 m (t.v.d) in TRU1-1 reflect lower stiffness
(Young's modulus ~3 GPa) at the measurement site, compared to the typical 11 GPa of the Opalinus Clay (Fig. 5).
This particular measurement was taken within a weak lens in the Opalinus Clay and is not accounted for by our
models due to the assumptions made while setting up the models. In general, stiffer formations such as the
Schwarzbach-Villigen formation, Zeglingen formation and the basement have broader stress ranges in the
measured data due to their statistically larger stiffness variability, while weaker formations like the Opalinus Clay
exhibit narrower, more consistent stress distributions. Moreover, stiffer layers shield the weaker layers above
and below, reducing stress variability in these formations. In short, Fig. 5 clearly indicates that the differences of
the profiles from all the models are smaller than the measurement errors, represented by the length of the
horizontal red and blue bars, and that the differences between the fault agnostic models and the REF model are
insignificant.
The AP and AP-H models yield identical results. This indicates that increasing model resolution would not
significantly improve stress predictions in our study and that the resolution of the AP model is already sufficient.
This rules out resolution effects within the ApplePy models on the predicted stress magnitudes with respect to
the REF model.

## 4.2 2D results along a cross-section

### 4.2.1 Spatial variation of horizontal differential stresses ($S_{Hmax}$-$S_{hmin}$)

Fig. 6 illustrates the spatial variation of horizontal differential stresses ($S_{Hmax}$–$S_{hmin}$) for all model realizations along
a W-E cross section and Fig. 7 illustrates the corresponding quantitative differences relative to the REF model,
along the same cross-section. In the cross-sections in Fig. 6, $S_{Hmax}$-$S_{hmin}$ visually appears consistent between
different model realizations, except near the contact surfaces where noticeable localized stress concentrations
in the REF model occur (Fig. 7). The contact surfaces are not included in the fault agnostic models (REF-NF, AP,
and AP-H), which explains the larger differences in differential stresses ($\Delta(S_{Hmax}$-$S_{hmin})$) observed. The $\Delta(S_{Hmax}$-
$S_{hmin})$ exceeds ±2 MPa within 100 m of the fault, technically the contact surfaces. Beyond approximately 200 m
from the contact surfaces, $\Delta(S_{Hmax}$-$S_{hmin})$ across all models become more similar to each other, and differences
relative to the REF model typically remain below ±0.4 MPa, less than the average widths of the measured stress
magnitude ranges shown in Fig. 5. As distance from the contact surfaces increases, the magnitude of the $\Delta(S_{Hmax}$-
$S_{hmin})$ differences rapidly decreases. It is important to note that variations in the stress field occurring over lateral
distances smaller than 60 m cannot be numerically resolved in our models, as the minimum lateral resolution is
about 60–80 m in the AP-H model and approximately 80–150 m in the other model realizations (Table 2).
In addition to the spatial proximity to contact surfaces, the variation of $S_{Hmax}$-$S_{hmin}$ depends on the stiffness of the
geomechanical units. In specific Mesozoic units characterized by lower stiffness, such as from the Wildegg Fm.
of the Malm Group to the Klettgau Fm. of the Keuper group, and the Kaiseraugst Fm. of the Muschelkalk group
in the order shown in Table 1, the $S_{Hmax}$-$S_{hmin}$ typically is < 3.5 MPa. In contrast, units with high stiffness can exhibit
$S_{Hmax}$-$S_{hmin}$ exceeding 7 MPa, such as in the «Felsenkalke» + «Massenkalk» and the Schwarzbach-Villigen Fm. of
the Malm group, Schinznach and Zeglingen Fm. of the Muschelkalk group and the Dinkelberg Fm., Weitenau Fm.
and Crystalline basement (Fig. 6, Table 1). This trend is expected, as lower stiffness materials accommodate
deformation more readily, resulting in lower differential stresses, whereas stiffer units resist deformation,
leading to higher differential stresses.



A particularly notable observation is that the differential stress near the Neuhausen fault remains relatively
comparable across all models when compared to the magnitude of differences in $S_{Hmax}$-$S_{hmin}$ at other contact
surfaces. Despite the Neuhausen fault being either fully removed or mechanically disabled via a high friction
coefficient, the differential stress pattern across the 50-meter offset between the footwall and the hanging wall
is well replicated in the AP and the AP-H models in Fig. 6. This is attributed to the abrupt contrast in mechanical
properties across the Neuhausen Fault (Fig. 3; Table 1), which effectively mimics the local stress response, even
in the absence of explicit fault representation.

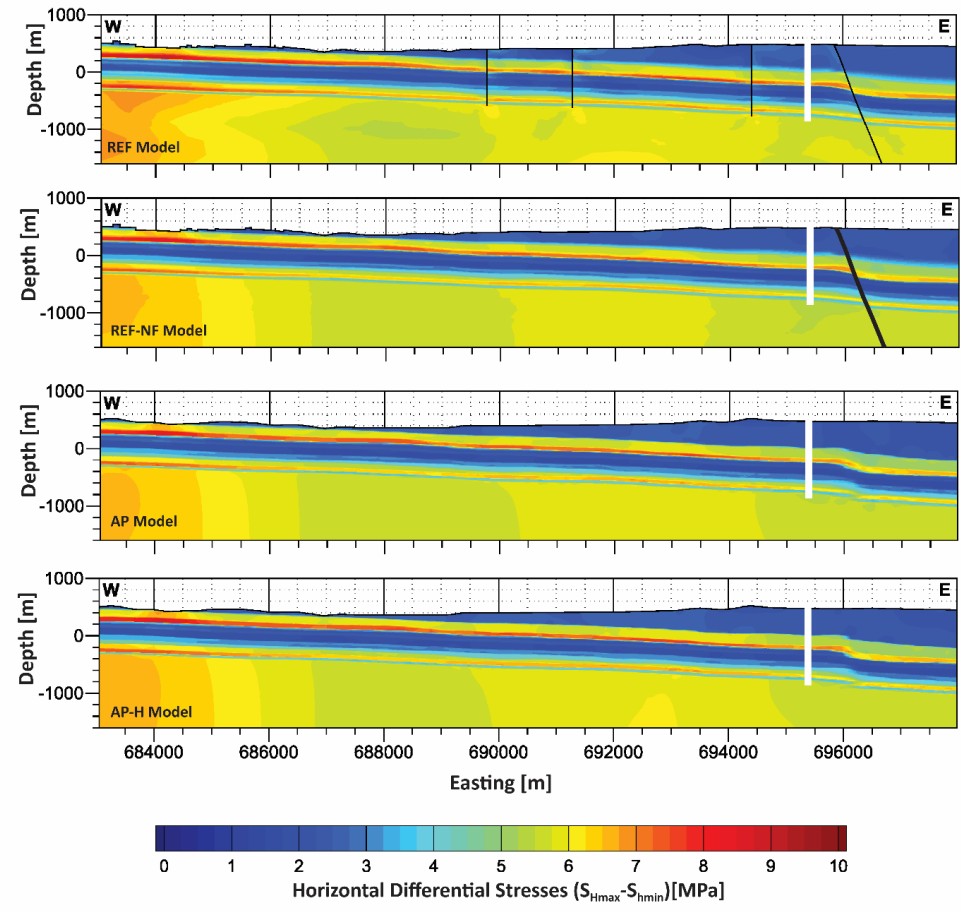


Figure 6: Comparison of the horizontal differential stresses ($S_{Hmax}$-$S_{hmin}$) along a W-E profile passing through TRU1-1 borehole (white blank space) and a fixed
Northing = 277548 m. The depths are referenced to the mean sea level (m.s.l). Higher $S_{Hmax}$-$S_{hmin}$ is observed in stiffer units whereas lower $S_{Hmax}$-$S_{hmin}$ are observed
in units with lower stiffness. The location of faults is indicated by black lines, similar to Fig. 3. In the REF-NF model, the thickness of the Neuhausen fault is
increased to signify that the fault has been mechanically deactivated by increasing the friction coefficient to 50, leading to no allowed slip/displacement along it.

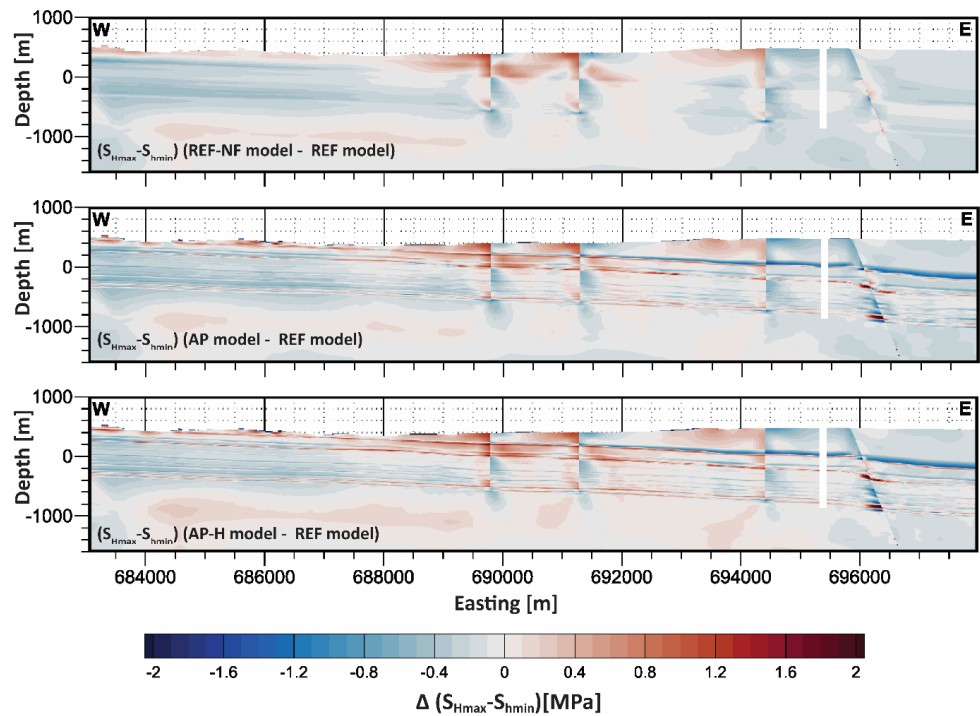


Figure 7: Comparison of the Δ (S$_{Hmax}$-S$_{hmin}$) between the models without faults and the REF model with active faults along the same cross-section as in Fig. 6. The slices show the difference with respect to the REF Model and are indicated at the bottom left of each slice. Key differences are primarily concentrated near contact surfaces within approximately 100 m. Although faults have not been directly indicated on the cross-sections, the location of the faults can be visually seen as sudden lateral discontinuities in an otherwise continuous change in Δ (S$_{Hmax}$-S$_{hmin}$). Visually, the individual geomechanical layers becomes more apparent in the two ApplePy models: AP and AP-H. This is due to the step-like transition between a higher-stiffness geomechanical unit and a lower stiffness geomechanical unit, leading to a more prominent visibility of stiffness contrasts at the geomechanical unit transitions.

## 4.2.2 Spatial variation of S$_{Hmax}$ orientation

In addition to S$_{Hmax}$-S$_{hmin}$, we also examined the S$_{Hmax}$ orientation (Fig. 8) and its variability along the same W-E cross section (Fig. 9). The largest S$_{Hmax}$ orientation variability is reoriented more within a distance of 100–200 m around the contact surfaces, similar to the observations of Δ(S$_{Hmax}$-S$_{hmin}$). At this distance, differences greater than 6° w.r.t the REF model are observed (Fig. 10). These differences tend to reduce to less than ±2° at lateral distances greater than 500 m from the contact surfaces. Within the near-field zone, which is < 300 m from the contact surfaces, stress concentrations are probably artifacts arising from numerical resolutions of the finite elements, which means that the values within 60–100 m from the contact surfaces should be interpreted with caution. Even under a hypothetical assumption that the observed variations are entirely fault-induced, S$_{Hmax}$ orientation changes are within 10° relative to the regional trend. Given that current stress indicator techniques cannot resolve S$_{Hmax}$ variations with a corresponding precision, these differences are not significant. Finally, increasing model resolution does not change our results, as seen when comparing the two ApplePy model results in Fig. 8 and Fig. 9.



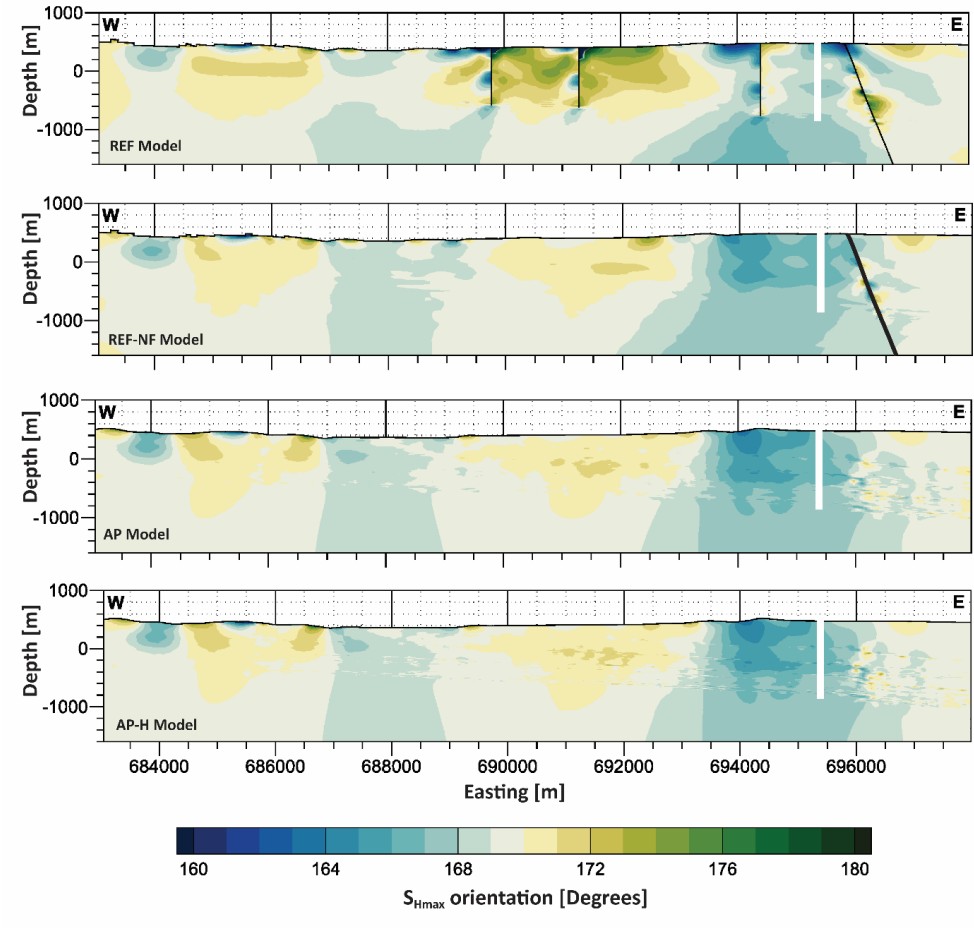


Figure 8: Comparison of the absolute $S_{Hmax}$ orientation along the same profile as in Fig. 6 and 7. The white black space indicates the TRU1-1 borehole. The location
of faults is indicated by black lines, similar to Fig. 3. In the REF-NF model, the thickness of the Neuhausen fault is increased to signify that the fault has been
mechanically deactivated by increasing the friction coefficient to 50, leading to no allowed slip/displacement along it.



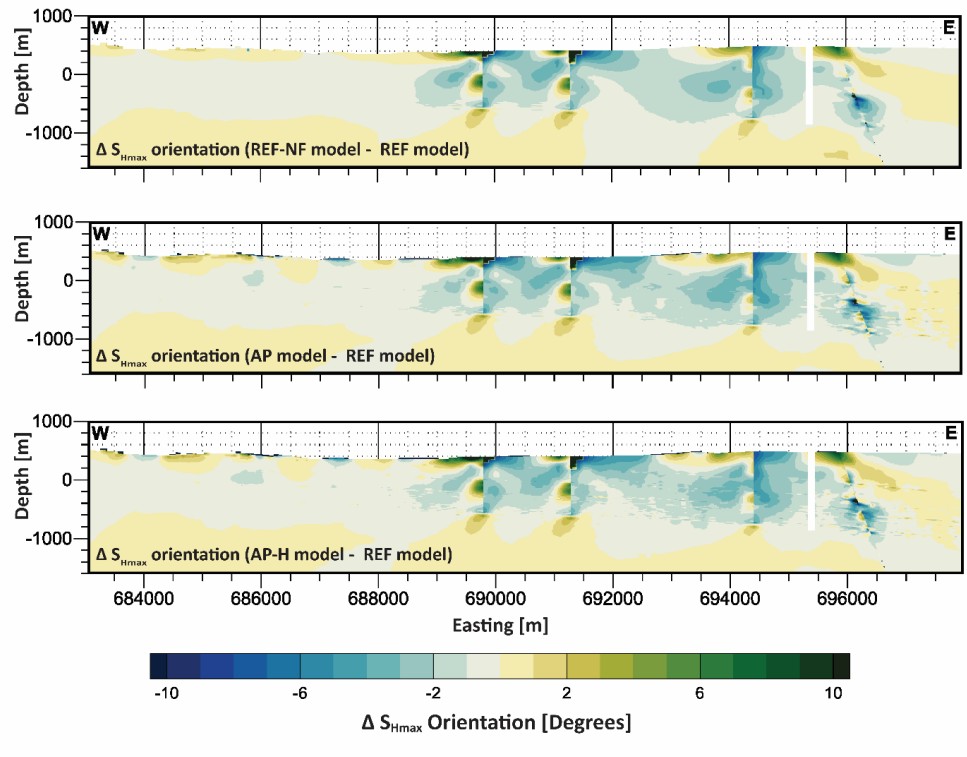


Figure 9: Comparison of the differences in $S_{Hmax}$ orientation with respect to the $S_{Hmax}$ orientation of the REF Model along the same profile as in Fig. 6. Key differences are primarily concentrated near contact surfaces within approximately 100-200 m. Note that the range of the color scale is smaller than the uncertainties of the stress orientation data records from the best stress indicator techniques i.e, Borehole Breakouts (BO) and Drilling induced tensile fractures (DITF).

## 4.3 Quantification of lateral extent of fault-induced stress changes.

To further investigate the spatial extent of fault impact on the stress state, we analyzed the lateral variation of stress tensor components by comparing results from different model realizations. For this purpose, stress values were extracted along a SW–NE oriented horizontal line located at a depth of 300 m (b.s.l). The horizontal line has been chosen such that it passes through as many fault structures as possible. The results of this comparison are presented in Fig. 10. To improve readability, the results from AP model were not plotted as it is clear from Fig. 5, 7 and 9 that AP and AP-H model results are almost identical.

The $S_{Hmax}$ and $S_{hmin}$ magnitudes of different model realizations largely overlap each other along the horizontal line (Fig. 10). A difference of ~0.5 MPa is observed in $S_{Hmax}$ magnitude and ~1 MPa is observed in the $S_{hmin}$ magnitudes between the REF-Model and the fault agnostic models, within ~500 m to the faults. However, these differences are less than the widths of the stress magnitude measurement ranges (Fig. 5). In general, the horizontal stress magnitudes from the REF model have an abrupt change in the vicinity of the faults, deviating from the continuous trend followed by other model realizations. The differences in the $S_{Hmax}$ magnitudes reduce to <0.2 MPa beyond a distance of about 500 m away from the fault. The differences in the $S_{hmin}$ magnitudes follow the same pattern as the $S_{Hmax}$ magnitude, and also reduce beyond a distance of about 500 m away from the fault.

Similarly, the $S_{Hmax}$ orientation of the REF model shows negligible deviations of <2° in the undisturbed rock volume, away from the faults and a deviation of 2°–6° up to 1 km from the modeled faults. According to the quality ranking scheme of the $S_{Hmax}$ orientation from the World Stress Map, the A-quality dataset, data of highest





quality, has an uncertainty of ±15° (Heidbach et al., 2025a). Considering this, the orientation deviations seen in
Fig. 10 are negligible and well below the uncertainties of the in situ indicators.
Near the Neuhausen fault, there is a localized abrupt change in the stress tensor components within ~100 m on
either side of the modelled fault for all the model realizations. An important observation is that this abrupt
change occurs not only in the REF model but also in the models without any faults. These stress changes are
primarily controlled by the lateral stiffness contrasts due to the offset and not by the mere presence of the faults.
Overall, the differences are < 0.2 MPa in stress magnitudes and < 2° in $S_{Hmax}$ orientations beyond 1 km from the
fault, which is far less than the uncertainties of the stress magnitude measurements. Even in a conservative
approach, it is clear that the effect of faults on the stress field is within about 1 km from the fault core. This
conclusion aligns with the findings by Reiter et al. (2024), who, through generic model studies, found that
significant stress changes due to faults only occur within a distance of few hundred meters, partly up to 1 km
next the fault.

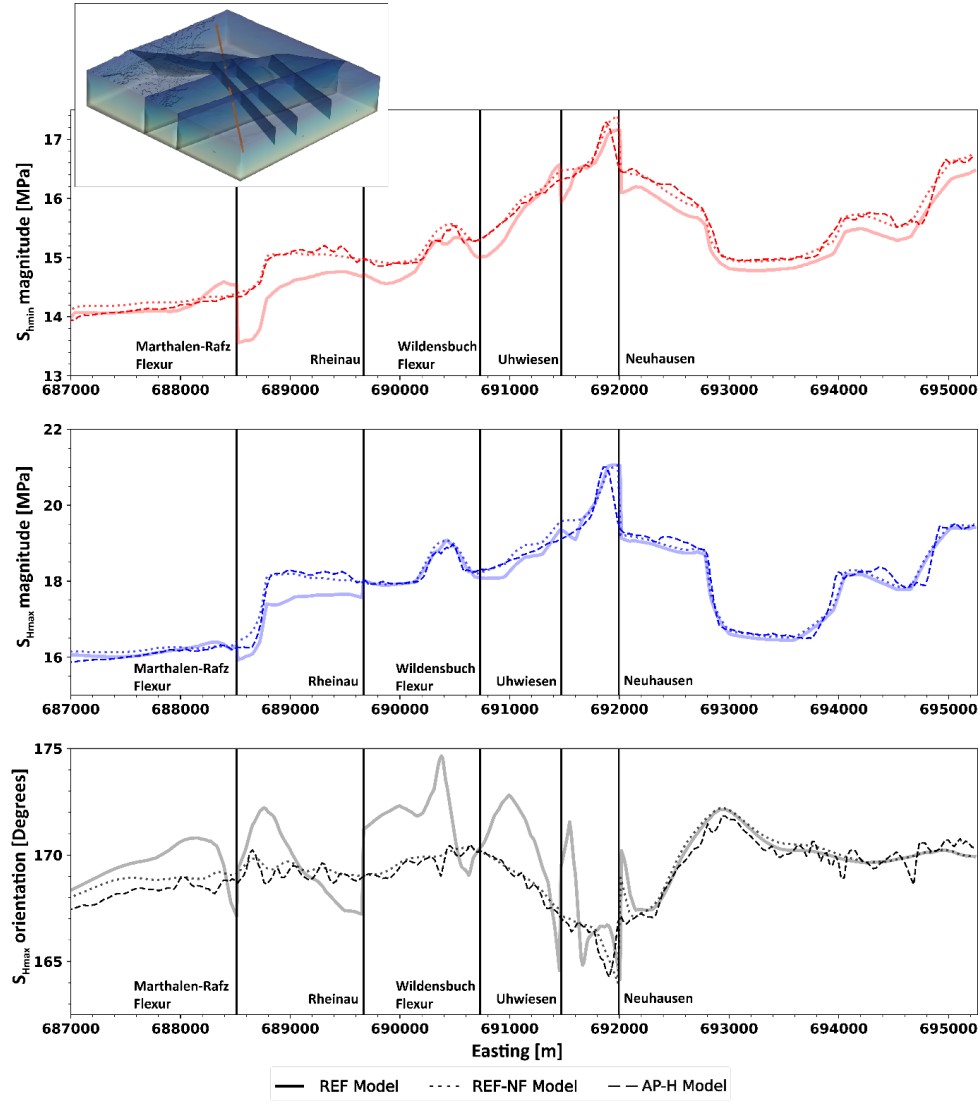




Figure 10: Effects of faults on Stress tensor components along a SW-NE horizontal line at 300 m (bsl). The location where the horizontal profile meets the modelled
faults are denoted by black vertical lines with the respective fault names. The red color lines represent the $S_{hmin}$ magnitude, blue represent the $S_{Hmax}$ magnitude
and black represent the $S_{Hmax}$ orientation. The line styles represent different model realizations. Note that the value of the y-axis is different for each sub-plot.
Notice that there is an abrupt change in the profile in all model realizations, at the Neuhausen fault, indicating that stress changes are caused by lateral stiffness
contrasts and not by the 'mere' fault presence. The location of the horizontal line, indicated in red, is shown in the 3D insert at the top left.
## 5. Discussion
### 5.1 Comparison with observed $S_{Hmax}$ orientation data
The orientation of the maximum horizontal stress ($S_{Hmax}$) is the most widely available component of the reduced
stress tensor. It is also the easiest component to analyze because it can be averaged and visualized with respect
to the fault on stress maps (Fig. 1). This topic was a subject of several earlier studies (Yale et al., 1993; Yale et al.,
1994; Yale and Ryan, 1994; Yale, 2003; Rajabi et al., 2017c; Heidbach et al., 2018). The $S_{Hmax}$ orientation can be
determined from different stress indicators, such as from direct borehole-based measurements, earthquake
focal mechanisms, geological indicators or passive seismic methods (Amadei and Stephansson, 1997; Zang and
Stephansson, 2010; Heidbach et al., 2025a). Among these, direct borehole-based data such as borehole
breakouts (BOs), drilling-induced tensile fractures (DITFs), and hydraulic fracturing (HFs) are commonly regarded
as the most reliable techniques (Bell, 1996a; Zang and Stephansson, 2010).
In the ZNO study region, 11 $S_{Hmax}$ orientation data records are available from HFs, DITFs, and BOs. The mean $S_{Hmax}$
orientation from these data is 170° with a standard deviation of ± 11°(Nagra, 2024c, b; Heidbach et al., 2025b).
The individual standard deviation of each data record is between ± 9° and ± 19° indicating that rotations smaller
than ± 11° cannot be resolved. As the differences between the REF model and the three realizations without
faults as displayed in Fig. 9 is smaller than ± 10°, the potential impact cannot be resolved with the stress indicator.
Furthermore, most of rotations observed are located in the near-field of the fault. At a distance of 1000 m from
the fault, the rotation is < ± 2° and thus clearly below the resolution limit.
The 1 km spatial distance limit can also be confirmed by viewing the $S_{Hmax}$ orientation from the boreholes in
correlation with their distance from the nearest faults. The TRU1-1 borehole is less than 1 km from the
Neuhausen fault. Similarly, the MAR1-1 and RHE1-1 boreholes are closest to the Rheinau fault. The average $S_{Hmax}$
orientation from the BO, DITF and HF is ~165° along the TRU1-1 borehole, ~175° along the MAR1-1 borehole and
~172.5° along the RHE1-1 borehole (Nagra, 2024b, c). Comparing the $S_{Hmax}$ orientation values from these three
boreholes to the regional $S_{Hmax}$ orientation value of 170° ± 11° already strengthens the argument that the faults
have minimal effects on $S_{Hmax}$ orientation even at a distance of less than 1 km.
### 5.2 Impact of varying fault friction coefficient of the implemented faults
In geomechanical modelling, the fault strength is commonly described by Mohr-Coulomb criteria and hence
characterized by its friction coefficient and cohesion (Brandes and Tanner, 2020). In most geological settings, the
friction coefficient varies between 0.6 and 1.0 in reservoirs with depths where normal stresses are < 200 MPa on
a pre-existing fracture plane (Byerlee, 1978; Zoback and Healy, 1984). In stark contrast, significantly lower
friction coefficient values are found in geological settings with extremely weak lithologies, overpressured fault
cores and in faults with very large offset and/or high slip rates (Morrow et al., 1982; Morrow et al., 1992; Di Toro
et al., 2011; Hergert et al., 2011; Li et al., 2022). Cohesion varies with different lithologies but for the pre-existing
faults, it is commonly assumed to be zero. In general, the value of friction coefficient values varies between 0.4
and 0.8, and is standardly taken as 0.65 (Hawkes et al., 2005; Kohli and Zoback, 2013). In northern Switzerland,
taking the lithology and the geological setting into consideration, the values of apparent fault friction coefficient
values commonly range from 0.6 to 1.0, and very rarely to 0.4 (Kastrup, 2002; Viganò et al., 2021). As seen in
the studies by Kastrup (2002), apparent fault friction values of 0.2 are extremely rare in the Switzerland and only
occur at depths more than 10 km.
We investigate the effect of varying friction coefficient of the contact surfaces on the predicted in situ stress
state and re-calibrating each model with different friction coefficient seperately . We consider all the realistically
possible values of friction coefficient in Switzerland but it must be kept in mind that the friction coefficient values
below 0.6 were categorized as 'anomalous' in Switzerland (Kastrup, 2002). The results of stress magnitudes and
orientation from friction coefficients 0.2, 0.4, 0.6 and 0.8 are compared to friction coefficient of 1.0, the value

hhh
dfg



we use in REF model (Fig. 11). We see that change in friction coefficient do not significantly affect our model
results beyond lateral distances of 1 km. Even within 1 km from the faults, both the horizontal stress magnitudes
have observable variations < 1 MPa and < 5° for the $S_{Hmax}$ orientation variations. These variations reduce to <
0.25 MPa in both minimum and maximum horizontal stresses, and < 2.5° in the $S_{Hmax}$ orientation beyond 1 km
from the faults. The maximum variations, still far less than the uncertainties in the measurements of the stress
magnitudes and resolvable $S_{Hmax}$ orientations, occur at a friction coefficient of 0.2. For the other values of friction
coefficient, the results are very much comparable to the REF Model, with friction coefficient of 1. This is to show
that changing the friction coefficient has a negligible effect on the predicted stresses in our model but might not
be the case in other studies.
These findings are in line with the results from the generic studies by Homberg et al. (1997) and Reiter et al.
(2024), who studied the impact of variable friction coefficient on different stress tensor components and found
that lower values of friction coefficient lead to a higher stress perturbation near the modelled fault. This is also
seen in Fig. 11 and is because of possible decoupling at the fault and consequently a better dissipation of stress
at the faults, facilitated by lower friction coefficients. The studies also showed that this effect is limited to a
distance of 1 km from the fault zone.

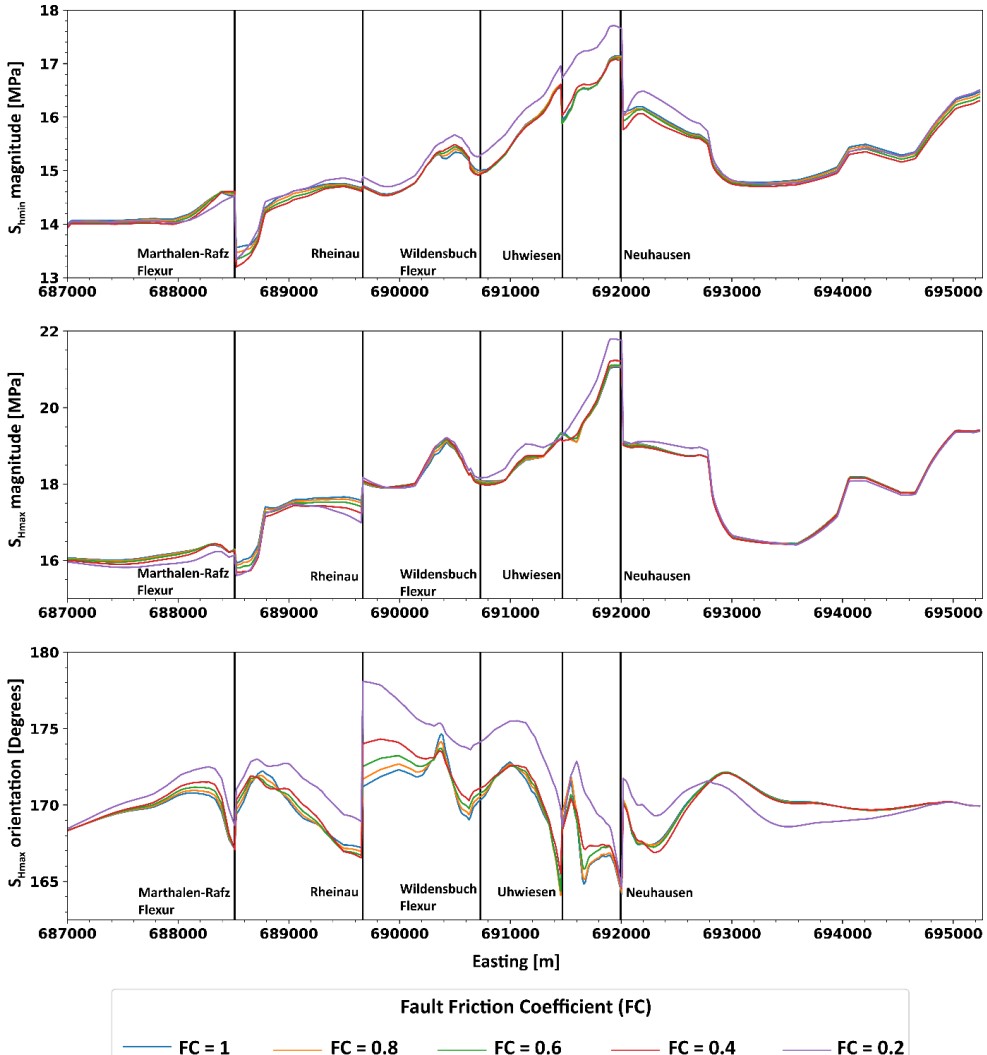




Figure 11: Impact of friction coefficient on the stress tensor components. The lower the friction coefficient, the larger the stress perturbations near the faults.
The results are plotted along the same horizontal line profile at the depth of -300 bsl, as used in Fig. 10. The stress perturbations increase with decrease in friction
coefficient values but their effect is observable only within ~1 km from the modelled faults. Given that the average width of the $S_{hmin}$ and $S_{Hmax}$ ranges are 0.7
MPa and 3.5 MPa respectively, the differences seen above are negligible.

## 5.3 Dependence of the modeling results on fault implementation

Faults in the REF model are represented as contact surfaces, a common and effective approach for large-scale
geomechanical simulations. Using contact elements to model faults seems to be a reasonable simplification for
large, field-scale reservoir models, where the actual width of the fault core is much smaller than the overall size
of the model. Hence, contact surfaces are computationally efficient for reservoir-scale models where actual fault
zone widths are negligible compared to model dimensions (Caine et al., 1996; Treffeisen and Henk, 2020). Since
our interest is on reservoir scale, alternative fault representation using e.g. continuous rectangular finite element
grid, or a continuous curvilinear finite element grid in a homogenized continuum (Henk, 2009, 2020) are not used
in our study. Furthermore, the results from Treffeisen and Henk (2020) and Reiter et al. (2024) show that the
stress and strain perturbations from different technical fault implementations vary only within a few tens to a
few hundred meters from the fault representation. As we focus only on the far-field stress state, it can be safely
assumed that the choice of fault implementation approach does not significantly affect the far-field results.
Although a numerical value does not exist for what is universally defined as far-field stresses, our model indicates
that at a distance of > 500 m from the faults the impact is clearly smaller than the uncertainty of the model itself
and smaller than the expected variability of the stress field (Nagra, 2024). As seen in Fig. 9, the influence of faults
on the stress field is limited to within 1 km from the contact surfaces. Beyond this distance, the choice of the
fault representation approach would not no significant impact on the predicted in situ stress state.

## 5.4 Limitations of the study's results and future outlook

In the REF model, the faults, represented by contact surfaces, are simplified and a unified representation of
numerous small fault patches that were interpreted from the 3D seismic interpretation. This simplification is
necessary for an easier and reasonable representation of fault structures and the consequent computational
simulation feasibility of the model. However, the reality is more complex. In the subsurface, faults often occur in
clusters and display heterogeneous geometry, composition and structure (Tanner and Brandes, 2020). Large
faults are often accompanied by zones of secondary faults, which can extend the spatial influence of faults on
the stress state. Small fault segments of the primary fault and the associated secondary faults can lead to a higher
stress concentration along the fault surfaces, complicating the interaction between faults and the in situ stresses
(Jones, 1988; Maerten et al., 2002). A single fault may also have complex geometry with multiple bends (Saucier
et al., 1992; Roche et al., 2021), increasing its influence on stresses compared to the planar faults.
Our study focusses on a reservoir scale, in the order of a few kms, to predict present-day stress variation in the
area of interest. While seven faults were implemented in the REF model, many more fractures or joints exist in
reality but cannot be resolved at our current lateral resolution of approximately 70–100 m and the available
structural geological data. Including these would significantly increase the element count and computational
demand, far beyond the scope or need of most studies. It is important to emphasize that the focus of the results
is only the far-field present day stresses, and in an intact and undisturbed rock volume.
Furthermore, extreme cases exist where large-scale faulting separated the crust into distinct fault blocks, each
having an independent $S_{Hmax}$ orientation between adjacent fault blocks of the same field (Yale et al., 1994; Yale
and Ryan, 1994; Bell, 1996b; Kattenhorn et al., 2000; Yale, 2003; Hergert and Heidbach, 2011; Hergert et al.,
2011; Li et al., 2019; Qin et al., 2024). But, as seen in our study area, if the Mesozoic sediments are not massively
faulted or fractured, and have sufficiently large differential stresses, and are located in an intraplate Foreland
Basin setting, it could be expected that the impact of faults on the stress state would only be within 1 km from
the fault zone. However, further investigation is needed for other geological settings, with different lithologies
such as salt domes, anhydrite or crystalline rock formations or regions where faults exhibit more complex
geometry with more curvature/ bends, or with extremely large total offsets and high slip rates to confirm broader
applicability of our results.



## 6. Conclusion

We evaluated the influence of faults on the regional stress state using 3D geomechanical models of the Zürich Nordost siting region, which are calibrated on a robust dataset of 30 minimum and 15 maximum horizontal stress magnitudes from two boreholes. We directly compare the predicted stress states between models where faults have been modelled as contact surfaces and models where faults have been excluded or mechanically deactivated. Our findings show that faults cause only local stress perturbations, within 500 m from the contact surfaces, with their impact becoming negligible beyond 1 km from the fault core. At this scale, stress variations are mainly controlled by contrasts in rock stiffness on the juxtaposed formations rather than just the relative mechanical weakness presented by the fault plane. The variations between the model realizations must also be viewed in conjunction with the rock stress variability, which in turn results from stiffness variability. The fault-induced stress effects at distances > 1 km are smaller than the typical resolution limits of stress data and uncertainties of the stress magnitude measurements, which is ±15° for $S_{Hmax}$ orientation and 0.7–3.5 MPa for stress magnitude, derived from the description of stress magnitudes as ranges. Importantly, omitting faults from the modeling workflow can reduce model-setup and computational time from months to 1–2 days using alternative discretization strategies, without sacrificing stress prediction accuracy. These findings provide valuable guidance for efficient and reliable reservoir-scale geomechanical modeling including repository site assessments, where predicting far-field in situ stresses in intact rock volumes is essential given that the storage sites are located away from active faults (>1km) in an intact and undisturbed rock volume.

## Author Contribution

LSARV: Conceptualization, Formal analysis, Methodology, Model preparation, Validation, Visualization, Writing (original draft preparation), and Writing (review and editing).

OH: Conceptualization, Data curation, Funding acquisition, Project administration, Resources, Supervision, Validation, and Writing (review and editing).

MZ: Resources, Software, Supervision, Validation, and Writing (review and editing).

KR: Methodology, Resources, Model preparation, Validation, and Writing (review and editing), Funding acquisition.

AH: Funding acquisition, Project administration, and Writing (review and editing).

MR: Conceptualization, Visualization, Writing (review and editing).

SBG: Resources, and Writing (review and editing).

TH: Visualization, Writing (review and editing).

## Competing Interests

The authors declare that they have no conflict of interest.

## Disclaimer

Publisher's note: Copernicus Publications remains neutral with regard to jurisdictional claims made in the text, published maps, institutional affiliations, or any other geographical representation in this paper. While Copernicus Publications makes every effort to include appropriate place names, the final responsibility lies with the authors.

## Acknowledgements

We thank NAGRA for providing access to the extensive dataset used in our study. We also thank SQuaRe and SpannEnd 2.0 for the funding.



## Financial Support

The authors gratefully acknowledge the funding provided by the Bundesministerium für Umwelt, Naturschutz,
nukleare Sicherheit und Verbraucherschutz through the project SQuaRe (project number: 02E12062B), and by
the Bundesgesellschaft für Endlagerung (BGE) through the project SpannEnD 2.0 (https://www.spannend-
projekt.de). Additional support was provided by the National Cooperative for the Disposal of Radioactive Waste
(Nagra), Switzerland.

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
