# Peer review of "Spatial influence of fault-related stress perturbations in northern"

_EGUsphere, 2025_

## Author Response (AR1)

Color explanation for preprint egusphere-2025-4559

Comments from referees

Author's response

Author's changes in manuscript: line in track changes file | revised manuscript | description

Other changes made in manuscript: Not suggested by reviewers.

**Reviewer 1: comment on 27 Oct 2025 and replied on 27 Nov 2025.**

Dear Dr. Beaudoin,

I have completed my review of "Spatial Influence of Fault-Related Stress Perturbations in Northern Switzerland" by L.S.A.R. Velagala and co-authors. Overall, I found this to be a well-written manuscript on a topic that is highly appropriate for inclusion in Solid Earth. As such, I recommend that it be accepted for publication after minor revision.

Here are several items that I think – once addressed by the authors – will further improve the manuscript:

With regards to the initial conditions of the numerical models, it is unclear if the maximum principal stress is vertical, signifying a "normal faulting stress state" because the text only discusses the magnitudes of the horizontal stresses. Adding the vertical stress curves to Figure 5 would be a simple way to provide clarification. Also, I think it would be helpful if the authors clearly indicated whether the stress magnitudes represent total or effective stresses.

In our geomechanical numerical models, the initial stress state is established by applying gravitational loading (producing the vertical stress, $S_v$) together with the lateral boundary conditions representing the regional tectonic stresses (producing additional horizontal stresses, $S_{Hmax}$ and $S_{hmin}$). In the initial conditions, we do not make any assumption that $S_v$ is the maximum principal stress ($S_1$). Instead, the three principal stresses ($S_1, S_2, S_3$) develop naturally from the applied loading and the lateral boundary conditions.

To further clarify, we provide a plot (attached below) showing the depth profiles of the $S_1, S_2, S_3$ together with $S_v$ along the Trüllikon borehole. The left panel displays the magnitudes of $S_1, S_2, S_3$ and $S_v$ as a function of depth, showing that $S_v$ is not always identical to the $S_1$. $S_v$ more closely corresponds to one of the principal stresses depending on the local stress regime, plotted as Regime Stress Ratio in the right panel [Simpson, 1997]. While not clearly visible in the plot, there is a very minute difference in between the magnitudes of the principal stresses and the vertical stress. We did not originally include vertical stress curves in Figure 5 because the differences between the model scenarios were small and only noticeable in the shallowest ~100 m. However, we agree that including them will improve clarity, and we will add the vertical stress profiles to the revised figure. We also clarify that all stresses presented in the manuscript refer to total stresses, as pore pressure effects were not considered in this study. We will state this explicitly in the revised manuscript.

[Figure]

Changes: 449 | 365 | Added vertical stress profiles.

152-153 | 124-125 | Added that the stresses in our study are total stresses.

Although the authors provide some text to address their choice to employ a linear elastic constitutive relationship (section 2.2.1), I think this area could benefit from additional discussion. While I agree with their assertion that this modeling assumption helps to simplify the analysis, many geomechanical modeling studies have shown that stress state evolution is impacted by inelastic deformations that occur when elastic limits are reached. As such, I think the authors could strengthen their arguments by including some text to demonstrate that their loading conditions lead to stress conditions that remain in the elastic domain.

The Mesozoic sedimentary units in the Northern Switzerland can be regarded as tectonically stable with low strain rates and are mechanically intact. This makes linear elastic constitutive relationship an appropriate first-order approximation for modeling the present-day stress state. Several geological factors, specific to the region and the siting area, support this assumption:

- No significant tectonic deformation since Miocene with extremely low strain rates of ~1-3 m/Myr/Km,
- Lack of active faulting at repository depths and no evidence of any quaternary faults' reactivation, and
- The differential stresses; S1-S3 of the units range between 0.5 MPa and 13 MPa), far below their measured uniaxial compressive strength limits (33-180 MPa) [Nagra, 2024]. Therefore, the units are far below their peak strength so they are safely within an elastic domain.

Due to these reasons, plastic yielding or damage is not expected under current stresses. Since the primary objective in this paper was to assess the far-field, first-order effects of fault-related stress perturbations at a safe distance (> one km) from faults, the assumption of linear elasticity is appropriate. We briefly mentioned about these region-specific details in the Section 5.4: 'Limitations of the study's results and future outlook'. However, the reviewer is correct that we could have explained this better and explicitly, and we will address this in the modified draft.

Changes: 263-272 | 221-230 | Reasons behind linear elastic constitutive relationship assumption.

The analysis of horizontal stresses (Figures 6-9) could be improved if the authors provided a few enlargements of key figures. For example, enlargements of the top portion of Figure 6 that illustrated the differential stress near the faults. And while the cross-sectional views are useful, I think that one (or more) horizontal (depth) slices would help the readers better appreciate the spatial variability in horizontal stress orientation and magnitude.

We agree that horizontal (depth) slices would offer a complementary perspective on the lateral spatial variability of the stress magnitudes and orientation. Therefore, we will include atleast one or two Figures from each model realization containing horizontal depth slices along the target storage zone, and a stiffer and mechanically contrasting unit above or below it.

While enlarging the regions near the faults from Figures 6–9 is technically possible, we found that such close-up views do not result in providing additional interpretive value to the reader. As discussed in Section 4.2: "2D results along a cross-section", the high localized stress concentrations adjacent to the faults are primarily numerical artefacts arising from element resolution limitations in particular near the fault tips. To show this, we provide a zoomed example of the area between Rheinau and D2 faults (below figure) from the top inset of Figure 7, originally showing $\Delta$ ($S_{Hmax}$-$S_{hmin}$) between the REF-NF and REF model. The enlarged view does not reveal information beyond what is already visible in the full image. For this reason, we propose to retain the original cross-section figures while adding the additional horizontal slices.

[Figure]

Changes: 545 and 582 | 405 and 430 | Added horizontal depth slices.

While the authors discuss their approach to representing the faults with contact interfaces and provide an analysis of the impact of different friction coefficients on horizontal stress magnitudes and orientations (Figure 10), it is unclear if actual slip occurs on any of the faults, and if so, what is the magnitude of slip. A brief discussion would be helpful for the readers.

The Neuhausen Fault is the only known active fault in our study area and is a tertiary-origin fault. In our numerical models, minor amounts of slip in the order of a few 10s of cm occur on this fault after the application of the lateral displacement boundary conditions (shortening of 4.1 m in the N–S direction and 0.82 m in the E–W direction to achieve the best-fit w.r.t. the stress magnitude data). This slip produces shear stress decrease along the fault that is small compared to the much larger background in situ stresses and the differential stress S1-S3. We will add a brief discussion to clarify that minor slip occurs but does not influence the overall stress field analysis.

Changes: 710-714 | 532-535 | Discussion about the magnitude of the slip.

Lastly, I think the manuscript's overall impact on the geomechanical interpretation of subsurface stress states and faulting would be enhanced if the authors could postulate how their results might differ if the scenario involved a different initial stress regime (i.e., strike-slip or thrusting faulting).

For low strain intraplate regions, we think that our findings are general and independent of the stress regime, but we cannot prove this with our model as it resembles a specific geological setting. Nevertheless, the differential stress S1-S3 is generally increasing with depth in the upper brittle crust (regardless of the stress regime) whereas the slip on the intraplate faults does not. Assuming that the slip-rate and thus, the stress changes along the faults are in the same order for different faulting styles, their impact on the in-situ stress is local and confined to the near-field of the fault. We will add our interpretation to the discussion in more detail.

Changes: 671-674 | 497-500 | influence of stress regime on stress perturbations.

779-784 | 590-596 | region-specific results.

In the end, we would like to thank the reviewer once again, as well as all others involved in the discussion phase, for taking the time to provide their valuable comments and suggestions. We are sincerely grateful.

**Concerned References**:

Simpson, R. W.: Quantifying Anderson's fault types, Journal of Geophysical Research: Solid Earth, 102, 17909-17919, https://doi.org/10.1029/97JB01274, 1997.

Nagra: In-Situ Stress Field in the Siting Regions Jura Ost, Nördlich Lägern and Zürich Nordost, NAGRA, Wettingen, Switzerland, NAGRA Arbeitbericht NAB 24-19, 131 pp., 2024.

**Reviewer 2: comment on 02 Dec 2025 and replied on 15 Dec 2025.**

Dear authors, dear Nicolas,

Please find below my review of the submitted MS "Spatial Influence of Fault Related Stress Perturbations in Northern Switzerland" by Velagala et al.

The MS presents the results of a 3D numerical modelling study targeting a restricted area north of Zurich, Switzerland. The authors employed a classical FE method with and without contact elements to simulate faults, the aim of the modelling being the simulation of potential stress perturbations, caused by the latter pre-existing discontinuities, and their quantification. It is concluded that the impact of the faults on stress orientations and magnitudes is negligible.

From the formal point of view, the text still contains typos, grammar mistakes and reads very often rather awkward (see some examples listed in "Minor comments" below). It is readable but needs to be seriously improved, which is somewhat puzzling, considering that it is stated in the paper that all the eight authors contributed to the writing. Also, some of the scientific/technical terms of the Ms are improperly used or, at least, ambiguous (e.g. "flexure", "stiffness", see comments below). In turn, the figures are informative and of excellent quality. I have, however some minor suggestions to improve them.

From the scientific point of view, the modelling study is a very classical one but it appears robust and its conclusions are sound. I would, however, raise some criticisms that I hope will help the authors to improve the Ms.

First of all, I would moderate the "universality" of the results of the study. For example, stress and structural data gathered in the vicinity of the San Andreas fault (Zoback et al. 1987) demonstrate that stress perturbations can be far more significant than what is advanced in the present Ms. The latter conclusion from Zoback et al. (1987) finds support in e.g. the paleostress reconstructions of Homberg et al. (1987), which evidence stress rotations of up to 20° near the Morez Fault Zone in the French Jura, the numerical modelling of Pascal and Gabrielsen (2001), later on benchmarked by the in-situ stress measurements of Myrvang and Roberts (2004), and the field observations and geomechanical modelling of Maerten et al. (2016). In brief, the results advanced in the present Ms are representative of the modelled area and cannot be safely extrapolated to other regions.

Pascal, C. & Gabrielsen, R.H. 2001: Numerical modelling of Cenozoic stress patterns in the Mid Norwegian Margin and the northern North Sea. Tectonics 20, 585-599.

Roberts, D. & Myrvang, A. 2004: Contemporary stress orientation features and horizontal stress in bedrock, Trøndelag, central Norway. Norges geologiske undersøkelse Bulletin 442, 53-63.

Zoback, M. D., M. Zoback, V. Mount, J. Suppe, J. Eaton, J. Healy, D. Oppenheimer, P. Reasonberg, L. Jones, C. Raleigh, I. Wong, 0. Scotti, C. Wentworth, New evidence on the state of stress of the San Andreas fault system, Science. 238. 1105-1111, 1987.

We agree with the reviewer that the results of our study apply in the strict sense only to Northern Switzerland and probably to similar geological settings in Foreland Basins. We will strengthen this perspective in Section 5.4 (Limitations of the study's results and future outlook), where we already addressed this point to some extent. We will rephrase the text accordingly in our revised manuscript to avoid any implication of universality. Before doing so, however, we would like to clarify how the results of our work relate to the studies cited in the comment, as this helps to contextualize the significance of our findings within the broader literature.

We acknowledge the significance of the studies cited by the reviewer and will integrate these in the discussion section of our manuscript. However, the fault systems discussed in these studies i.e the Morez Fault Zone (Homberg et al., 1997), the Møre-Trøndelag Fault Complex (Pascal and Gabrielsen, 2001; Roberts and Myrvang, 2004), and the San Andreas Fault (Zoback et al., 1987) differ significantly from the Alpine Foreland Basin in terms of their governing tectonic processes and fault/fault-system structure complexity. In particular, the San Andreas Fault is a major plate boundary structure, while the Møre-Trøndelag Fault Complex is a complex and large crustal

fault system in crystalline rock. Both show 100s of meters to kilometers of relative displacement whereas the maximum observed fault displacement in our model is less than 50 meters.

Furthermore, the applied friction coefficients of μ = 0 in the DEM model of Pascal and Gabrielsen (2001) to reproduce the 'shield effect' of the Møre-Trøndelag Fault Complex, and the friction values of μ < 0.2 derived for the San Andreas fault from heat flow measurements and numerical models (D'alessio et al., 2006), are not comparable with those in the Northern Switzerland (Kastrup, 2002). Also, the low differential stresses reported in Pascal and Gabrielsen (2001), which facilitates larger and large-scale rotations of $S_{Hmax}$ due to local and regional stress perturbations (Ziegler et al., 2017), are not observed in the Zürich Nordost region.

In summary, the studies cited by the reviewer address settings at plate-boundary and major crustal fault systems characterized by high-strain, low-friction faults, and low differential stresses. Our work is in intraplate setting in the Alpine Foreland Basin and focuses on the impact of relatively small faults on an otherwise undisturbed rock volume, which are based on the 3D seismic interpretations from the region. We will strengthen the manuscript wording to reflect this, as suggested by the reviewer.

Changes: 779-784 | 590-596 | region-specific results.

The origin of the data used to calibrate the models and summarised in Table 1 is enigmatic. Please add the apparently missing references and explain briefly how these data were obtained. Also, important information is missing in the text about the model set-up (e.g. dimensions, type of elements used to mesh the models).

We will add the missing reference for the dataset summarized in Table 1. While we had stated the general procedure used to measure the datasets in the manuscript, we agree that the readers would benefit from a brief description of the procedures used to obtain these datasets and the model set-up, and will add this information in the revised manuscript. We will edit Section 2.2 to make this information more readily visible to the readers.

Changes: 293 | 249 | Missing reference added.

291-293 | 247-249 | How the data was obtained.

154 and 193 | 126 and 164 | Model dimensions.

283-284 | 240-241 | Types of elements used to mesh the model.

The modelling considers faults as contacts, whereas natural faults are complex zones of deformation, where rock rheology can be drastically different from that of the "intact" host rock and, thus, fault zones may promote pronounced stress perturbations. A thorough discussion on the latter is crucially missing in the paper. Also, the models do not involve fault tips, whereas it is well-known that they host the most dramatic stress perturbations (e.g. Homberg et al. 1997). Once again, the latter point should be addressed in the discussion section.

We agree that simplifying the faults to a contact surface in our models does not capture the real nature and complexity of faults on the 1-10 m scale. This is acknowledged in the original manuscript in Section 5.3 (Dependence of the modeling results on fault implementation) and Section 5.4 (Limitations of the study). However, we justify our approach in the following and will add these arguments in the model set-up section.

Given that our models have lateral dimensions of 14.7 × 14.8 km$^2$ the width of the fault core with a few 10s of meters is small compared to our model dimensions and the numerical resolution. Thus, we cannot resolve the internal fault structures. Furthermore, the semi-generic study of Reiter et al. (2024), which uses a model of similar dimension, assumptions, data and geometrical setting shows that different technical implementations of faults do not have an impact on the modelled stress tensor components in the far-field, i.e. at distances larger than a few hundred meters. This is also supported by other works (Caine et al., 1996; Treffeisen and Henk, 2020) and was briefly mentioned in our Section 5.3 (Dependence of the modeling results on fault implementation). However, we agree that this information must be better expressed in our manuscript, and we will do this in our revised manuscript.

Regarding the work of Homberg et al. (1997) and Nicol et al. (2020), we acknowledge their findings that there are large stress rotation at fault tips, but they note that these rotations are localized and rarely extends beyond a few hundred meters from the fault tip, which is in general agreement with our work. Furthermore, we also have fault tips in our model where the contact surfaces end within the Mesozoic sediments (and they actually show stress concentrations as well as larger stress rotations at the tips). However, an analysis of the stress state at the actual physical fault tips of these faults falls outside the scope of our study and would need a much higher numerical resolution around the fault tips. To give a more complete picture about the fault tips stresses, we will expand the text in the revised manuscript.

Changes: 732-743 | 548-559 | Reasons behind modelling faults as contact surfaces.

767-773 | 583-589 | Role of fault tips in stress perturbations.

Major comments

L24: "Coulomb friction". The term does not apply to contact surfaces as Coulomb friction (i.e. internal friction) is a property of the continuous medium that controls rupture (i.e. creation of shear fracture not sliding along a pre-existing one). I recommend to use "Byerlee friction" or "Amontons friction".

We thank the reviewer for the careful attention to the used terminology. However, we respectfully disagree with the statement that the term 'Coulomb friction' does not apply to contact surfaces. Coulomb's law of friction describes the frictional resistance between two dry solid surfaces, where the frictional force is proportional to the normal force and independent of the apparent contact area (described by Amontons-Coulomb law). In fault mechanics, this directly applied to frictional sliding along pre-existing discontinuities.

It is true that in the Mohr–Coulomb constitutive framework, the friction coefficient is combined with a cohesion term to describe the strength of intact material, described as continuous by the reviewer. However, the same is also commonly used to describe the damaged or frictional sliding regime, in which the friction coefficient represents resistance to sliding along contact surfaces rather than internal friction of an intact medium. This usage is standard in rock mechanics. By contrast, Byerlee friction refers to an empirical relationship derived from the laboratory experiments on intact rock samples. Therefore, it represents a specific observed constraint on the friction coefficients for certain lithologies and at certain conditions, rather than a general definition of the frictional behavior. As the result, very low friction coefficients cannot be justified by the Byerlee's law alone. The term Coulomb friction has also been used to describe sliding along a pre-existing fracture by Zoback (2007) and Jaeger et al. (2007).

To avoid this confusion and to improve clarity for the readers, we will refer to μ as contact surfaces with a coefficient of friction, without using the term 'Coulomb', in the abstract and throughout the revised manuscript, as we will cite Zoback (2007) to explain the reasoning behind the proposed referencing of μ.

Changes: Throughout the manuscript | Throughout the manuscript | The term Coulomb's friction has been changed to friction coefficient throughout the revised manuscript.

L194: "flexures". The term "flexure" is traditionally used to indicate a mechanical process and not a geological structure. Please explain what do you mean by "flexure" (fold hinge? Drape fold?). Also, it is unclear to me how the "flexures" impact the stress field (e.g. if they are open folds I do not expect much of it). Please, justify their inclusion into the model.

The Marthalen-Rafz Flexur and Wildensbuch Flexur are monoclines (Nagra, 2024). These Flexures dominate the overlying Mesozoic strata in the siting area by a step-like bending rather than a discrete break in otherwise gently dipping strata. During the interpretation of 3D Seismic sections, several hundred fault patches were identified and for the model combined to seven individual faults and implemented as contact surfaces. Although, except for the Neuhausen fault which shows a distinctive displacement, the other faults do not show a measurable displacement in the footwall and the hanging wall, they are included in the model because they represent first-order geological structures (discontinuities in the otherwise reasonably continuous lithology) observed within the siting region. They also help in quantifying the maximum possible stress perturbations that such observed

structures could impose on the far-field stress state (Seen in Fig. 10). We will add brief clarifying information about these distinctions in the revised manuscript.

Changes: 230-235 | 192-196| Clarification about using the term 'Flexures'.

L199-200: "Neuhausen is the only fault that displays a stratigraphic offset". The statement is very puzzling, if the other faults do not display any offset then… they are not faults. Please, clarify in text.

We will clarify this in the revised manuscript.

Changes: 233-235 | 195-196| Inclusion of other faults in the model.

L241-242: "The mechanical properties E [GPa], v [-], and ρ [kg/m³], used in the models are derived from core tests and petrophysical logs obtained from the TRU1-1 and MAR1-1 boreholes." Please add references and indicate briefly what kind of "core tests" have been conducted.

We will clarify this in the revised manuscript, along with the references.

Changes: 291-293 | 247-249| Reference added and clarification of the tests provided.

Table 1: Please indicate also lithologies, formation names are not informative enough for the common reader.

We will add this information to the table.

Changes: 307 | 257| Information about the lithology added to the table.

"3.1 Model discretization Strategies". Please, explain briefly in introduction the rationale of the modelling and why different models are tested. As it is now, the text is rather confusing and does not allow the reader to grasp right away the adopted modelling strategy and its goals.

We agree that the text can be a little confusing to the readers. We would make the sentence clearer. While we had already mentioned in the manuscript that different models were tested with different element property assignment strategies to see if a simpler and faster python-based method would still provide the same results as the reference model, we will make this information better accessible in the revised manuscript.

Changes: 335-341 | 279-285 | Why different models are tested.

L340 (and elsewhere in the Ms): "stiffness". The word "stiffness" has a precise meaning in continuum mechanics and involves typical dimensions of the object under consideration. Obviously, the term is erroneously used in the Ms. I presume that one has to read "Young's modulus" instead. Please correct.

The term stiffness contrast has been used in multiple earlier works by different authors to describe the contrast of the Young's modulus from one lithology to another. We would prefer to stay with this wording given that earlier publications and models of similar setting use the same terminology. However, we will introduce this choice in the beginning of the revised manuscript that stiffness contrast is similar to Young's modulus contrast.

Changes: 256-257 | 214-215 | Clarification regarding usage of the word Stiffness.

L397: "resulting in lower differential stresses". Rigorously speaking: "resulting in lower stress magnitudes" and potentially in lower differential stresses.

We agree with the comment. While "lower stress magnitudes" is more rigorous, we propose to retain the reference to differential stresses because this is the most relevant quantity for the processes discussed, and the link between stress magnitude and differential stress is implicit in this context.

Changes: No changes made | No changes made | No changes made.

L472: "Stress tensor components". Stress orientation is not considered to be a "stress component" traditionally. Please correct.

We will correct this in the revised manuscript. The term component has been changed to characteristic throughout the manuscript.

Changes: 67-68 | 56-57 | Change of terminology.

"5.2 Impact of varying fault friction coefficient of the implemented faults". This section reads like a presentation of some of the results and not like a discussion. Please consider moving it to section 4.

In the section 5.2 'Impact of varying fault coefficient of the implemented faults', we artificially lower the friction coefficient of the Reference model upto 0.2. This was not a part of the results section as we had to tweak the realistic parameters to extremely low values to explore the effect of friction coefficient on the predicted far-field stress state. Therefore, we propose that the subsection is better placed in the Discussion section than the Results section.

Changes: No changes made | No changes made | No changes made.

L518: "possible values of friction coefficient in Switzerland". Is there a particular reason to believe that physics operate otherwise in Switzerland compared to the rest of the universe? I presume that friction coefficients are there similar to the ones measured elsewhere on the planet. Please, rephrase.

We agree to clarify the sentence and will change the text accordingly.

Changes: 692-696 | 517-520 | Clarification added.

 Minor comments

L98: "Fig. 1D shows". Should be called after Fig. 1C (has not been called before).

We acknowledge that figure panels are typically called in sequential order. However, we intentionally refer to Fig. 1D later in the introduction to emphasize the reservoir scale, where the main knowledge gap addressed by this study lies. We believe this improves the conceptual flow of the introduction.

Changes: No changes made | No changes made | No changes made.

L132: "these three scales". Please be specific. What scales do you mean?

We believe that the sentence is redundant and have removed it from the revised manuscript. The Figure is well-explained without this sentence.

Changes: No changes made | No changes made | No changes made.

L133 (and somewhere else in the text): "reduced stress tensor". Please define, there is no unique form for it.

We will clarify this in the revised manuscript.

Changes: 67-68 | 56-57 | The form of the reduced stress tensor has been defined.

Fig. 2: please, in order to help the reader indicate the cross section presented in Fig. 3.

We will indicate this in the revised manuscript.

Changes: 209 | 180 | The cross-section has been indicated on Fig. 2

L189: "the displacement applied" should read "the applied displacement".

We believe that the sentence is redundant and have removed it from the revised manuscript. The Figure is well-explained without this sentence.

Changes: Sentence deleted | Sentence deleted | Sentence deleted.

L213: "two key simplifying assumptions". I presume one should read "two simplifying key assumptions".

We will address this comment in the revised draft as they pertain to use of English grammar by making use of a scientifically accepted tool 'Grammarly' to assist us with making grammatical changes.

Changes: No changes made | No changes made | No changes made.

L261 "Fig. 5". Please call Fig. 4 before calling Fig. 5.

We acknowledge that it is a common norm to call figures in alphabetical order. However, the manuscript is better served by calling Fig. 5 before Fig. 4 due to the information flow.

Changes: No changes made | No changes made | No changes made.

L271: "vertical displacement is constrained". I find the expression rather ambiguous (i.e. it could be constrained to e.g. 5 mm/yr or to accelerate progressively or…). I presume you mean something like "zero vertical displacement is prescribed".

We will make this change in the revised manuscript.

Changes: 326-327 | 272 | Clarification added.

Fig. 4: please add scale.

This conceptual figure is well served and scale-independent. It represents a visual comparison of the two strategies used in our work for model discretization and mechanical property assignment to the geomechanical units, and the scale is immaterial for the comparison.

Changes: No changes made | No changes made | No changes made.

L300: "faster by approximately an order". Please be specific and add numbers.

We believe that the sentence is redundant and have removed it from the revised manuscript. The Figure is well-explained without this sentence.

Changes: Sentence deleted | Sentence deleted | Sentence deleted.

Table 2: what is the reason for (slightly) changing boundary conditions from one model to the other?

The slight changes in the values of the boundary conditions between the different model realizations are explained by the Reference model having contact surfaces. The contact surfaces allow dissipation of elastic energy by means of displacement which is not possible in the fault agnostic models. It is important to note that each model used in our study are calibrated to the same dataset to get the best fit. We will explain this in the revised manuscript.

Changes: 397-399 | 325-327 | Clarification added about the changing boundary conditions.

L336-337: "vertical red line changing with depth". Unclear, please rewrite.

We believe that the sentence is redundant and have removed it from the revised manuscript. The Figure is well-explained without this sentence.

Changes: Sentence deleted | Sentence deleted | Sentence deleted.

Fig. 6: the figure is redundant as Fig. 7 provides already all needed information (same comment concerning Fig. 8, Fig. 9 is sufficient). Please consider removing Figs. 6 and 8.

We acknowledge that Figures 6 and 7, and Figure 8 and 9 provides similar information. However, a goal of our paper is also to show that the differences between the model realizations are smaller than the unresolvable uncertainties in the stress data. This valuable piece of information is not seen in Figure 6 and 8, therefore we propose still keeping Figure 7 and 9.

Changes: No changes made | No changes made | No changes made.

**Concerned References**:

Caine, J. S., Evans, J. P., and Forster, C. B.: Fault zone architecture and permeability structure, Geology, 24, 1025–1028, https://doi.org/10.1130/0091-7613(1996)024%3C1025:FZAAPS%3E2.3.CO;2, 1996.

d'Alessio, M. A., Williams, C. F., and Bürgmann, R.: Frictional strength heterogeneity and surface heat flow: Implications for the strength of the creeping San Andreas fault, Journal of Geophysical Research: Solid Earth, 111, https://doi.org/10.1029/2005JB003780, 2006.

Homberg, C., Hu, J. C., Angelier, J., Bergerat, F., and Lacombe, O.: Characterization of stress perturbations near major fault zones: insights from 2-D distinct-element numerical modelling and field studies (Jura mountains), Journal of Structural Geology, 19, 703–718, https://doi.org/10.1016/S0191-8141(96)00104-6, 1997.

Jaeger, J. C., Cook, N. G. W., and Zimmerman, R. W.: Fundamentals of Rock Mechanics, 4, Blackwell Publishing, 2007.

Kastrup, U.: Seismotectonics and Stress Field Variations in Switzerland, Dissertation, Swiss Federal Institute of Technology Zurich (ETH Zurich), Zurich, 162 pp., https://doi.org/10.3929/ethz-a-004423062, 2002.

Maerten, L., Maerten, F., Lejri, M., and Gillespie, P.: Geomechanical paleostress inversion using fracture data, Journal of Structural Geology, 89, 197–213, https://doi.org/10.1016/j.jsg.2016.06.007, 2016.

Nagra: Geosynthesis of Northern Switzerland, NAGRA, Wettingen, Switzerland, NAGRA Technischer Bericht NTB 24-17, 604 pp., 2024.

Nicol, A., Walsh, J., Childs, C., and Manzocchi, T.: Chapter 6 - The growth of faults, in: Understanding Faults, edited by: Tanner, D., and Brandes, C., Elsevier, 221–255, https://doi.org/10.1016/B978-0-12-815985-9.00006-0, 2020.

Pascal, C. and Gabrielsen, R. H.: Numerical modeling of Cenozoic stress patterns in the mid-Norwegian margin and the northern North Sea, Tectonics, 20, 585–599, https://doi.org/10.1029/2001TC900007, 2001.

Reiter, K., O. Heidbach, and M. O. Ziegler (2024), Impact of faults on the remote stress state, Solid Earth, 15(2), 305–327, doi:10.5194/se-15-305-2024.

Roberts, D. and Myrvang, A.: Contemporary stress orientation features and horizontal stress in bedrock, Trøndelag, central Norway, NGU Bull., 442, 53–63, 2004.

Treffeisen, T. and Henk, A.: Representation of faults in reservoir-scale geomechanical finite element models – A comparison of different modelling approaches, Journal of Structural Geology, 131, 103931, https://doi.org/10.1016/j.jsg.2019.103931, 2020.

Ziegler, M. O., O. Heidbach, A. Zang, P. Martínez-Garzón, and M. Bohnhoff (2017), Estimation of the differential stress from the stress rotation angle in low permeable rock, Geophysical Research Letters, 44(13), 6761–6770, doi:10.1002/2017GL073598.Zoback, M. D.: Reservoir Geomechanics, Cambridge University Press, Cambridge, https://doi.org/10.1017/CBO9780511586477, 2007.

Zoback, M. D., Zoback, M. L., Mount, V. S., Suppe, J., Eaton, J. P., Healy, J. H., Oppenheimer, D., Reasenberg, P., Jones, L., Raleigh, C. B., Wong, I. G., Scotti, O., and Wentworth, C.: New Evidence on the State of Stress of the San Andreas Fault System, Science, 238, 1105–1111, https://doi.org/10.1126/science.238.4830.1105, 1987.

**Other changes made in the manuscript: Not suggested by reviewers**

All the figures have been edited for more clarity and to comply with the size requirements of the EGU- Solid Earth journal, i.e., below 5 mb.

All headings and subheadings have been edited to comply with the requirements of the EGU- Solid Earth journal, i.e., sentence style capitalization.